

# How subsurface and double-core anticyclones intensify the winter mixed layer deepening in the Mediterranean sea

Alexandre Barboni[1,2], Solange Coadou-Chaventon[3], Alexandre Stegner[1], Briac Le Vu[1], and Franck Dumas[2,4]

[1]Laboratoire de Météorologie Dynamique, IPSL, Ecole Polytechnique, 91128 Palaiseau, France
[2]Département de Recherche en Océanographie Physique, Service Hydrographique et Océanographique de la Marine (SHOM), 13, rue du Chatellier 29200 Brest, France
[3]Département de Géosciences, Ecole normale supérieure, 24 rue Lhomond, 75005 Paris, France
[4]Laboratoire d'Océanographie Physique et Spatiale, Equipe côtière, rue Dumont Durville, 29280 Plouzané

**Correspondence:** Alexandre Barboni (alexandre.barboni@lmd.ipsl.fr)

**Abstract.** The mixed layer is the uppermost layer of the ocean, driven by atmospheric fluxes. It follows a strong seasonal cycle, deepening in winter due to buoyancy loss, shoaling very close to the surface with summer restratification. Recently global and regional studies showed a mixed layer depth (MLD) modulation by mesoscale eddies with the seasonal cycle. In winter, MLD tends to be deeper inside anticyclonic eddies and shallower inside cyclonic ones. Several studies proposed a scaling

law with eddy sea surface height deviation. However they were done globally or regionally with eddy composites mostly representative of surface-intensified structures and using monthly averaged climatologies as reference. The Mediterranean sea contains a wide variety of mesoscale eddies, with the specific presence of several large anticyclones living up to 4 years, in particular in the Eastern basin. These anticyclones were surveyed over the past decade with numerous Argo floats deployments. Several floats remained trapped inside anticyclones for months and enabled to record 16 continuous MLD time series inside

13 long-lived anticyclones at a fine temporal scale on the order of the week. MLD evolution at anticyclone cores reveals a stronger winter deepening, reaching sometimes deeper than 300m, compared to always less than 100m in the neighboring background. MLD evolution also does not coincide inside- and outside-eddy, starting to restratify outside of eddies, while it keeps deepening and cooling MLD at anticyclone core for a longer time. We then bring to light a restratification delay of one month on average between the anticyclones and their background, sometimes reaching more than 2 months. Extreme MLD

anomalies of up to 330m that would be smoothed in composite analyses can then be observed when the winter mixed layer connects with a preexisting subsurface anticyclonic core, greatly accelerating mixed layer deepening. On the opposite, the winter MLD sometimes does not achieve such connection but homogenizes a second subsurface layer, then forming a double-core anticyclone with spring restratification. Formation of several double-core anticyclones in the Eastern Mediterranean is accurately described in time. MLD restratification delay and connection with preexisting subsurface anomalies appear to be

determinant in MLD modulation by mesoscale eddy and highlights the importance of interaction with eddy vertical structure.



## 1 Introduction

The mixed layer corresponds to the ocean surface layer over which water properties are kept uniform through active mixing. It connects the atmosphere to the subsurface ocean through air-sea fluxes of heat, freshwater or other chemical components such as carbon (Takahashi et al., 2009; Large and Yeager, 2012). The mixed layer depth (MLD) controls how deep the mixing is acting, bringing water properties from below to the surface and the other way around. This depth is subject to pronounced seasonal variations, the mixed layer deepening with winter heat loss, while spring surface heating restratifies the column and the mixed layer gets shallower. Due to its importance for both ocean physics and biogeochemistry, global MLD climatologies were computed (de Boyer Montégut et al., 2004; Holte et al., 2017). Several MLD climatologies were also computed for the Mediterranean sea (d'Ortenzio et al., 2005; Houpert et al., 2015), showing specific dynamics in winter convective regions such as the Gulf of Lion, the Egean and the Adriatic seas or the Rhodes gyre, with biological impacts on plankton bloom (d'Ortenzio and Ribera d'Alcalà, 2009; Lavigne et al., 2013). However large spread in MLD was also observed in regions hosting intense anticyclones such as the Algerian, Ionian and Levantine basins (Houpert et al., 2015), highlighting the need to take into account the local impact of mesoscale eddies.

Recent development of automatic eddy tracking algorithms and eddy atlases (for example Chelton et al. (2011), Pegliasco et al. (2022) at global or Stegner and Briac (2020) at regional scale), combined with an increase of in-situ measurements thanks to the development of autonomous platforms (Le Traon, 2013), recently allowed to study the influence of mesoscale oceanic eddies on the MLD. It is now well known that anticyclonic (respectively cyclonic) eddies tend to deepen (shoal) the MLD (Dufois et al., 2016; Hausmann et al., 2017; Gaube et al., 2019). Eddies actually amplify the MLD seasonal cycle, the deepest MLD anomaly being reached during winter (Hausmann et al., 2017; Gaube et al., 2019). A first mechanism was proposed by Williams (1988), the eddy-modulation of the MLD being related to their induced Sea Surface Temperature anomaly (SSTA). Indeed, anticyclonic (cyclonic) eddies are usually associated to positive (negative) SSTA (Hausmann and Czaja, 2012) (this is at least true in winter in the Mediterranean sea, see Moschos et al. (2022)), leading to stronger (weaker) heat loss during the winter and triggering enhanced (reduced) ocean convection and therefore deeper (shallower) MLD. In addition Hausmann et al. (2017) in the Southern Ocean and Gaube et al. (2019) for the global ocean found out that the eddy MLD anomaly, computed from eddy composites, scales with the eddy SSH amplitude. Gaube et al. (2019) proposed the same linear trend at the global scale $\pm 1m$ MLD anomaly for $1cm$ SSH for both cyclones and anticyclones. Physical drivers controlling the eddy-induced MLD are supported by other studies showing an eddy-modulation of air-sea exchanges. Villas Bôas et al. (2015) found ocean heat loss enhanced (respectively reduced) in anticyclones (cyclones) in energetic regions of the South Atlantic ocean, once again scaling with eddy amplitude, for both sensible and latent heat flux. Frenger et al. (2013) showed enhanced rainfall and cloud cover above anticyclones in the Southern ocean as a consequence of enhanced turbulent heat fluxes, but suggested a scaling with the eddy SSTA. Such relation should remain coherent, as Hausmann and Czaja (2012) also found anticyclone warm (cyclone cold) eddy SSTA to scale with the eddy amplitude in the Gulf Stream region. Altogether, eddy MLD anomalies are expected to be easily inferred provided that background measurements outside eddies are available, a promising link for





remote sensing application.

      However all these studies used coarse monthly temporal resolution, whereas mixed layer is driven by air-ocean fluxes and
      thus is expected to react at timescale close to the inertial period (D'Asaro, 1985; Lévy et al., 2012). If several studies showed
      the MLD and in particular the restratification to vary over timescales of a week at regional scales (Lacour et al. (2019) in the
North Atlantic or d'Ortenzio et al. (2021) in the Rhodes gyre), no one studied in details the temporal evolution of eddy MLD
      anomaly. A second limit in previous studies is the use of composite datasets that smooth out the wide range of non-linearities
      induced by regional eddies. If the composite analysis can provide a first order trend, this is likely not sufficient to quantify
      accurately the various impacts of wide diversity of individual eddies varying in size and intensity. A third - but linked - limit
      explicitly pointed out by Villas Bôas et al. (2015) and Hausmann et al. (2017) is their focus on surface-intensified eddies with
the most coherent surface signature. Indeed the relation between eddy-SSTA and SSH amplitude strongly relies on the hypoth-
      esis of a surface-intensified structure and Assassi et al. (2016) showed that it should not be the case for subsurface anticyclone
      due to isopycnals and isotherms doming above the eddy core. A more detailed comparison with more observational data is then
      lacking.

Several key ingredients make the Mediterranean sea an interesting region to study eddy influence on MLD. Due to repeated
      oceanographic campaigns and a continuous deployment of Argo profiling floats, the density of in-situ measurements is rela-
      tively high inside long-lived mesoscale eddies. Besides, the variety of mesoscale structures offers a wide range of dynamics,
      from intense Algerian and Ierapetra eddies needing cyclogeostrophic corrections (Ioannou et al., 2019) to subsurface eddy
      with strong density anomalies but weak SSH signature (Hayes et al., 2011). Moutin and Prieur (2012) also showed the vertical
structure, in temperature and salinity, of mesoscale eddies to be very different from one basin to another. Barboni et al. (2021)
      showed the marked subsurface difference between a new anticyclone detached from the coast compared to an offshore structure
      having been tracked more than a year. All these structures should have a different impact on the mixed layer. Another impor-
      tant parameter for mesoscale dynamics in the Mediterranean sea is the strong asymmetry between cyclones and anticyclones,
      remarkable in lifetime difference (Mkhinini et al., 2014). The deformation radius in the Mediterranean sea is indeed about 8 to
12 km (Kurkin et al., 2020), and cyclones are less stable when greater than the deformation radius and more subject to external
      shear (Arai and Yamagata, 1994; Graves et al., 2006). This leads to cyclones being predominantly below the effective reso-
      lution of SSH products about 20km (Stegner et al., 2021). As a consequence anticyclones are coherent larger vortices, while
      cyclones in the Mediterranean sea as detected by altimetry are instead cyclonic gyres bounded by topography or hydrographic
      fronts such as the Ligurian, South-Western Crete or Rhodes gyre (Millot and Taupier-Letage, 2005; Stegner et al., 2021). MLD
evolution inside these cyclonic gyres was already surveyed because of their importance for biological production, in particular
      with the development of BGC-Argo (d'Ortenzio et al., 2021; Taillandier et al., 2022). Apart for specific campaigns, Mediter-
      ranean anticyclones remain poorly analyzed despite being more coherent, and statistical comparison based on vertical profiles
      is lacking, with the noticeable exception of the BOUM campaign surveying 3 anticyclones across the basin in 2008 (Moutin





and Prieur, 2012).


This paper aims to study the temporal evolution of the mixed layer inside a wide diversity of long-lived anticyclones in the Mediterranean sea compared to the evolution of the background MLD. The goal is to quantify more precisely the local impacts of individual eddies on the winter mixed layer deepening. The paper is organized as follows. Section 2 describes the in-situ profiles database and the eddy detection and tracking algorithm. Section 3 details the methodology used to compute the MLD,

colocalize profiles and eddies in order to quantify accurately the MLD anomalies induced by individual eddies. In the section 4 we analyze the MLD evolution at anticyclone cores, provide statistical analysis over the variety of structures surveyed and discuss the impact of complex vertical eddy structure on winter mixed layer deepening. Finally in the last section 5 we discuss the possible physical drivers and implication for these MLD anomalies.

## 2 Data

### 2.1 In-situ profiles

A climatological database is created collecting in-situ profiles from the Coriolis Ocean Dataset for Reanalysis (CORA). Delayed-time (CORA-DT, Szekely et al. (2019b)) profiles are recovered from 2000 to 2019 (113486 profiles), and near-real-time (Copernicus-NRT, Copernicus (2021)) profiles are recovered from 2000 to 2021 included using the **history** release (22821 profiles). In addition, some profiles are not yet released in CORA-DT but available in Copernicus-NRT. This happens in partic-

ular when the salinity sensor of an Argo float has abnormal values but the temperature is still correct (by visual inspection and correct quality flag). As the MLD computation can be performed on a temperature profile alone, profiles were also retrieved in NRT mode after careful check, described in Appendix A. This provides an extra 20746 profiles from 2000 to 2019. Complete database then accounts for 157053 profiles in total.

### 2.2 Anticyclone detections : the DYNED Atlas

Eddy detections are provided by the Angular Momentum Eddy Detection and Tracking Algorithm (AMEDA). AMEDA is a mixed velocity-altimetry approach, using primarily a velocity field to compute streamlines and identifying possible eddy centers computed as maxima of local normalized angular momentum (Le Vu et al., 2018). From 1 January 2000 to 31 December 2019, AMEDA is applied on AVISO sea surface height (SSH) delayed-time product at a resolution of 1/8°with daily output. From 1 January 2020 to 31 December 2021, AMEDA is applied on AVISO SSH near-real-time day+6 product (Pujol, 2021),

at the same spatial and temporal resolution. In each eddy single observation (one eddy observed one day), AMEDA gives a center and two contours. The 'maximal speed' contour is the enclosed streamline with maximal speed (i.e. in the geostrophic approximation, with maximal SSH gradient) ; it is assumed to be the limit of the eddy core region where water parcels are trapped. The 'end' contour is the outermost closed SSH contour surrounding the eddy center and the maximal speed contour ; it is assumed to be the area of the eddy footprint, larger than just its core but still influenced by the eddy shear (Le Vu et al., 2018).





AMEDA gathers eddy observations in eddy tracks, allowing to follow the same structure in time and space, sometimes over
        several months. The eddy tracks collection in the whole Mediterranean sea constitutes the DYNED Atlas database (Stegner and
        Briac, 2020), and is available online (for the years 2000 to 2019) at : https://www1.lmd.polytechnique.fr/dyned/. From 2000 to
        2021, a total of 7038 (respectively 8890 ) anticyclonic (cyclonic) eddy tracks were retrieved. The asymmetry in eddy numbers
        is driven by a lifetime difference, anticyclones living noticeably longer, an asymmetry even more marked in the Levantine
basin (Barboni et al., 2021).

## 3    Methods

### 3.1    MLD computation

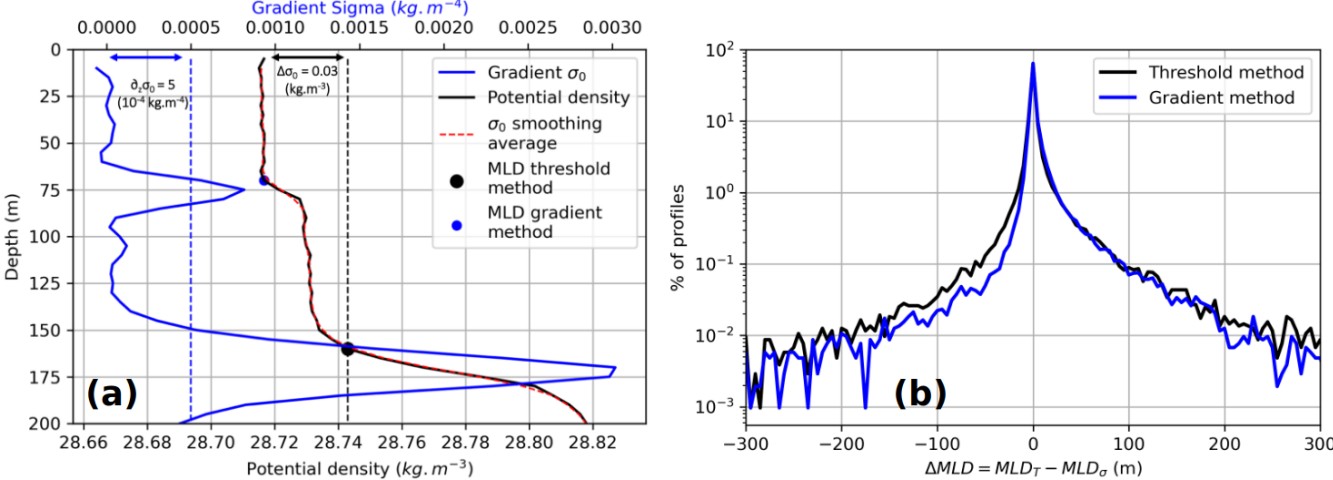

**Figure 1.** (a) MLD detection on one potential density profile with our gradient method (blue dot) and threshold method (black dot) and (b)
fraction of the profiles as a function of $\Delta MLD$ for the threshold (black line) and gradient (blue line) method

        The global analysis conducted by de Boyer Montégut et al. (2004) led to MLD detected by threshold values of $0.03 kg.m^{-3}$
        for density and $0.2°C$ for temperature, based on a reference depth at 10 m to avoid diurnal heating at the surface. In the Mediter-
ranean sea d'Ortenzio et al. (2005) used this methodology for a 0.5°resolution MLD climatology. Houpert et al. (2015) updated
        this climatology with 8 supplementary years of data, but opted for a 0.1°C temperature threshold. This more restrictive crite-
        rion enables to reduce the difference between the MLD based on temperature profiles with the one made on density profiles.
        Gradient methods are looking in a similar way for critical gradients as an indicator of the mixed layer base. Typical gradient
        threshold values in use are $2.5 \times 10^{-2} °Cm^{-1}$ for temperature profiles and range from $5 \times 10^{-4}$ to $5 \times 10^{-2} kg.m^{-4}$ for poten-
tial density profiles (Dong et al., 2008). Mixed gradient and threshold methods were also developed (Holte and Talley, 2009).



Here we aim to capture as accurately as possible the MLD evolution, which can vary on timescale shorter than a month. More specifically we observed in several cases that the threshold method (with criteria $\Delta\sigma = 0.03\ kg.m^{-3}$ and $\Delta T = 0.1$ °C) can miss the mixed layer and return the main thermocline instead (see Fig.1a)). This is characterized by a small jump in potential density (or in temperature) but a significant peak in the gradient profile, and happens mostly in the spring, probably due to a

start of restratification quickly mixed. We then chose a gradient method, including a first step using a threshold method. Using the following thresholds : $\Delta\sigma = 0.03\ kg.m^{-3}$ and $\Delta T = 0.1$ °C, we derive a first estimate of the MLD. If it is shallower than 20 m, we take it as our estimate of the MLD. Otherwise, we apply a three-point running average to remove small vertical-scale spikes and compute the gradient using a second-order centered difference. From the subsurface (20 m) up to the first MLD estimate, we apply a gradient method with the given gradient thresholds : $\partial_z\sigma = 5 \times 10^{-4}\ kg.m^{-4}$ or $\partial_z T = 2.5\ °C.m^{-1}$. If

the gradient fails to exceed the threshold within the given depth range, than the first MLD estimate is kept.

Threshold and gradient methods are limited by their dependence on the criterion values which can have strong influence on the MLD estimate. The relatively low gradient thresholds chosen here appeared to be necessary to catch the MLD in some of our profiles as higher thresholds would return the main thermocline (see Fig.1a). A sample of 400 randomly-picked profiles

colocated inside eddies was used for validation. We took a dataset of profiles inside eddies as our main interest is to get the seasonal evolution of the MLD inside eddies and wrong detection on double gradient profiles were found to be quite large inside eddies, exceeding 100 m sometimes. On this 400 profiles, 22 (5.5 %) of them were double gradient profiles using the threshold method on density profiles. This means that for these 22 profiles the MLD was overestimated with this method. Switching to the gradient method results in only 2 (0.5 %) double gradient profiles (with wrong detection). However, with the

gradient method comes some issues on profiles with small vertical-scale spikes despite the smoothing applied. For 2 profiles, the gradient method returned wrong MLD detection where the threshold method was correct. However, overall, the gradient method was found to be more accurate on estimating the MLD.

Moreover we aim that the chosen thresholds return similar estimates between a MLD obtained from temperature profile and

one from a potential density profile. Potential density is a better estimate of the stability of a layer, and thus MLD computed on potential density profiles should give a more reliable value. However, salinity (and hence density) suffers from data holes, representing about 15 % in our dataset. Temperature profiles then offer a good alternative in evaluating the MLD providing they give similar estimates that the ones obtained on potential density profiles. MLD is then computed on the density profile, and on the temperature is no density is available. The histogram of the MLD estimates difference is shown in Fig. 1b: the

gradient method appears to slightly reduce this difference with 64 % of the profiles leading to the same estimate and 94 % to less than a 30 m difference compare to 62 and 93 % respectively for the threshold method.

## 3.2 Eddy colocalisation and background estimate

In order to characterize the impact of anticyclonic eddies on the MLD seasonal evolution and spatial gradient, we need to accurately colocalize in situ profiles with eddy observations. However due to altimetric product interpolation and disparate



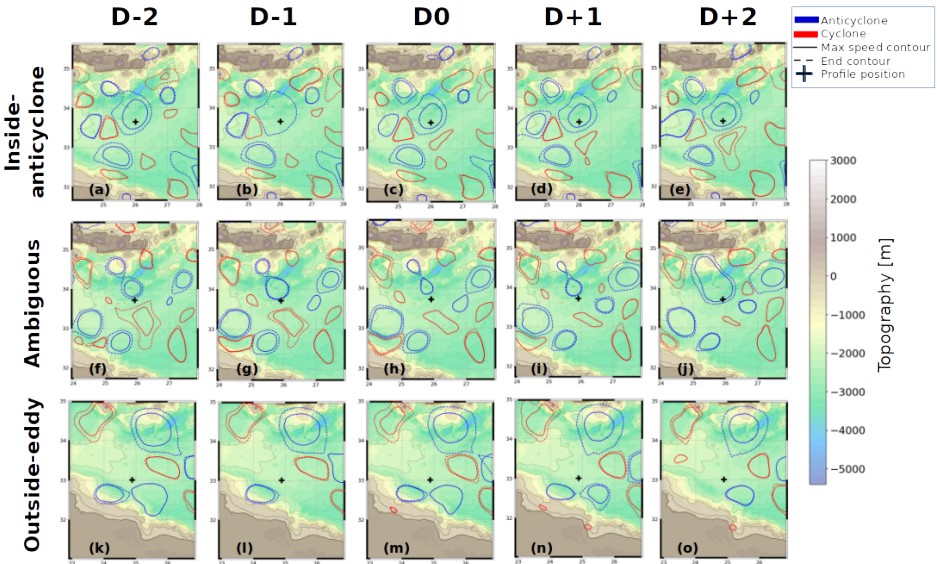

**Figure 2.** Profiles colocalisation with eddy contours for an 'inside-anticyclone' profile (panels (a) to (e)), an 'ambiguous' one (panels (f) to (j)) and an 'outside-eddy' one (panels (k) to (o)). Profile cast position is assumed fixed and compared to eddy contours at D-2, D-1, D, D+1, D+2, D being the profile cast date.

satellite tracks, SSH-based contours can vary a lot in size and position, making a single eddy observation less reliable in the Mediterranean sea (Amores et al., 2019; Stegner et al., 2021). Therefore we colocalize eddy observation and in situ profile at $\pm 2$ days. Assuming a profile position fixed at cast date D, it is then labeled as 'inside-eddy' if it remains inside the maximal speed contours of the same eddy at D-2, D-1, D0, D+1 and D+2 (at least 4 contours out of 5). This 4-out-of-5 threshold avoids to neglect a colocated profile when the eddy contour is not available for just one day (see Fig.2b). For the same purpose, here-

after the eddy center and the distance of a profile to the eddy center are averaged at $\pm 2$ days.

AMEDA also gives for each observation the last closed SSH contour (see section 2.2), inside which there is still an impact of the eddy shear, but outside of the maximal speed contour the water particles are not assumed to be trapped. The area between the maximal speed and last closed SSH contours are then considered as an intermediate zone to be discarded. Consistently with

the 'inside-eddy' definition, we label as 'outside-eddy' only profiles staying outside any eddy contours at $\pm 2$ days of its cast date. Any profile being neither 'inside-' or 'outside-eddy' is considered as ambiguous and discarded from this analysis. From 2000 to 2021, out of 157053 profiles retrieved in the Mediterranean sea, 104787 are labelled 'outside-eddy', 7939 are 'inside-anticyclone', 14919 'inside-cyclone', and the remaining 29410 'ambiguous' profiles are discarded from this analysis. This asymmetry between anticyclones and cyclones sampling is also due to heterogeneous oceanographic surveys (Houpert et al.,

2015), in particular the numerous glider missions in the Gulf of Lion, a cyclonic gyre with no large anticyclones (Millot and Taupier-Letage, 2005). Figure 2 illustrates the colocalisation method detailed above with 3 examples : an 'inside-anticyclone'





profile (Fig.2a-e), an 'ambiguous' one (Fig.2f-j) and an 'outside-eddy' one (Fig.2k-o). For this particular 'inside-anticyclone' profile, the maximal speed contour was missing at day D-1, but available the other days, and the profile was indeed cast close to the eddy center.


To follow the accurate evolution of the MLD inside anticyclones, we need a reference for comparison: an unperturbed, local and time-coincident ocean state without eddies, hereafter called 'background'. This outside-eddy background differs from a classic climatology used in previous studies (Gaube et al., 2019) by removing the eddy mean effect and by avoiding as much as possible time-averaging. The background of an eddy, at a given time $t$ and center position $C(t)$, is then constituted by the

mean of all profiles labelled as outside-eddy closer than 250km from $C(t)$, cast within $\Delta day = \pm 10$ days of the same year, or the previous or the following year ($\Delta y = \pm 1$ year). For example when computing the corresponding background of an eddy around 15 February 2018, the background encompasses profiles labelled 'outside-eddy', closer than 250km and cast from 5 to 25 February 2017, 2018 and 2019. A threshold on the number of profiles is required : if less than 10 profiles meet the distance, time and outside-eddy requirements, then no background is computed. At last we define as 'Background MLD' the median

MLD of the profiles constituting the background. Computing the median is preferred to the mean as the MLD distribution is not centered, being only towards the bottom. This computation is performed for each timestep, with temporal resolution of 5 days. As shown in Appendix Fig.A1, with the test case of the Ierapetra anticyclone over 2 years (corresponding events 'IER1-2' on Table1 and Fig.10), the background MLD is not highly sensitive to the choice of $\Delta day$ and $\Delta y$. The background MLD evolution is indeed similar with $\Delta day = 10, 15$ or $20$ days and $\Delta y = 0, 1$ or $2$ years. It is however important not to

take all years, as interannual variability then starts to smooth the background MLD evolution. On the other hand taking only profiles of the same year ($\Delta y = 0$) sometimes translates in not enough profiles to have a background estimate (see Fig.A1a). We therefore chose 10 days and 1 year as day and year intervals in order to capture MLD variation as short as possible, which is crucial for parameters varying quickly such as the MLD. For the two earliest recorded events ('MM1' in 2006 and 'MM2' in 2008, see Table1), $\Delta y$ is set to 2 years because no background MLD was available otherwise. Choosing $\Delta y = 1$ allows to have

accurate eddy induced anomalies without being corrupted by interannual variability of temperature and salinity fields, which can be marked in the Mediterranean, in particular in the Eastern basin (Ozer et al., 2017) and with a significant warming trend (Parras-Berrocal et al., 2020).

### 3.3 MLD evolution function fit

To describe more objectively the MLD seasonal evolution in the background, we performed a function fit using the Python

optimization routine `scipy.optimize.curve_fit`. MLD data points are selected per 5 days time steps. Background MLD is fitted by a skewed Gaussian, $t_{max}$ being the time when deepest MLD ($MLD_{max}$) is reached, $\sigma$ and $\tau$ respectively the





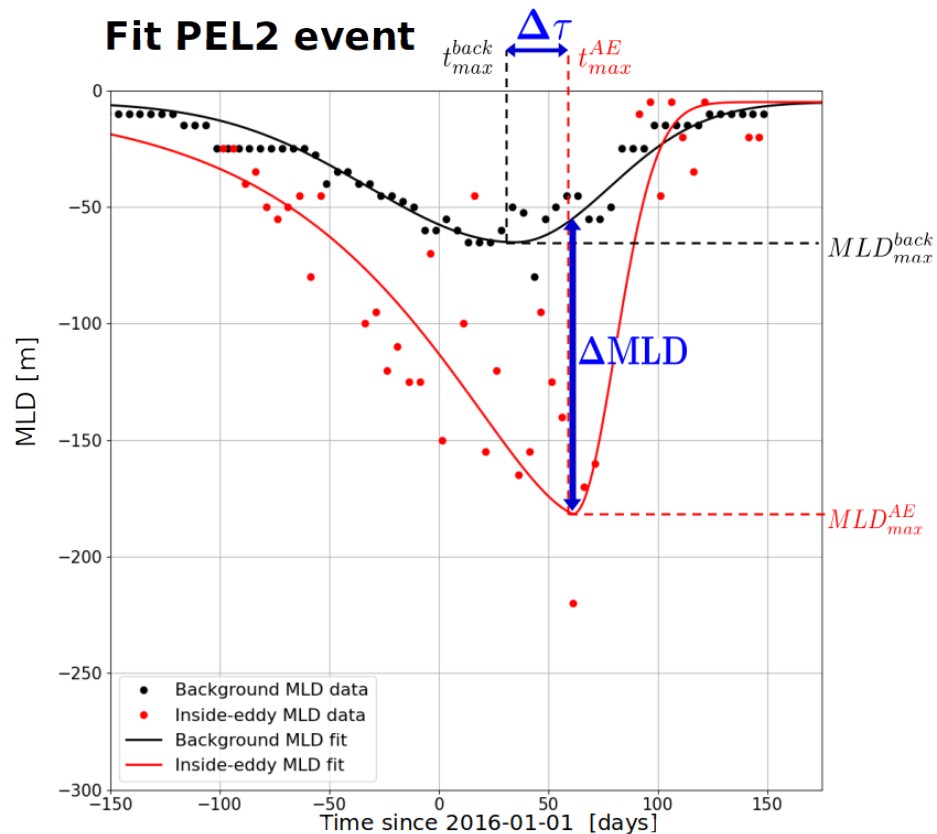

**Figure 3.** Detail of winter deepening event PEL2 in 2016 (see Table1 for details). Anticyclonic core MLD data are shown as red dots, background MLD as black dots, with time steps of 5 days. MLD fit is shown as a red line for the anticyclonic core (see Eq.2) and as a black line for background MLD (see Eq.1).

deepening and restratification timescales :

$$
\begin{aligned}
f(x) &= MLD_{max}^{back} \exp\left(-\frac{(t - t_{max}^{back})^2}{2\tau^2}\right) && \text{if } t > t_{max}^{back} \\
f(x) &= MLD_{max}^{back} \exp\left(-\frac{(t - t_{max}^{back})^2}{2\sigma^2}\right) && \text{otherwise}
\end{aligned}
\tag{1}
$$

This fit captures the background MLD evolution, somehow smooth, typically with a sharper restratification than deepening
$(\tau < \sigma)$. However this is not sufficient for the anticyclonic core MLD evolution that can have more abrupt variations, then
calling for a more complex fit with two deepening timescales $\sigma_1$ and $\sigma_2$ :

$$
\begin{aligned}
f(x) &= \left(MLD_{max}^{AE} - B\right) \exp\left(\frac{t - t_{max}^{AE}}{\tau_1}\right) + B \exp\left(\frac{t - t_{max}^{AE}}{\tau_2}\right) && \text{if } t > t_{max}^{AE} \\
f(x) &= MLD_{max} \exp\left(-\frac{(t - t_{max}^{AE})^2}{2\sigma^2}\right) && \text{otherwise}
\end{aligned}
\tag{2}
$$



To fit accurately the MLD evolution, and in particular the maximal depth reached, data are fitted with weights proportional to their depth. Because it is difficult to have long and continuous time series, data are often missing on the previous or next

summers. To ensure physical behavior, fit is forced back to 10m on the edges, miming summer stratification. The MLD anomaly ($MLD^{anom}$) is defined as the difference between the fitted background and anticyclonic core MLD. $MLD^{anom}$ is a function of time but reaches its maximum ($\Delta MLD$) at almost the same time than the absolute anticyclonic core MLD, as the latter has more amplitude than the background one. At last an advantage of the `scipy.optimize.curve_fit` routine is to provide the parameter covariance matrix, and hence an error estimate taking the square root of the covariance matrix diagonal

(Bevington et al., 1993). It can happen for the covariance matrix to have very large values, in this case we used an upper uncertainty of $\pm30m$ for $\Delta MLD$ and $\pm20$ days for $t_{max}^{AE}$. A fit illustration is provided in Fig.3 for the PEL2 event in 2016 (see later Table1), with dots as the real MLD data and fits in continuous lines, background in black and anticyclonic core in red. Using the fit routine, maximal MLD anomaly is then estimated for this event to $\Delta MLD = 127 \pm 13\,m$. On can also notice an absence of coincidence between deepest inside-eddy and background MLDs. Following previous notation, we can then define

a restratification delay of the anticyclonic core MLD, used throughout this study : $\Delta \tau = t_{max}^{AE} - t_{max}^{back}$. In the example shown in Fig.3, $\Delta \tau = 26 \pm 11$ days.

## 4 Results

### 4.1 Winter deepening connecting preexisting subsurface core

In order to investigate the relation between the anticyclonic core MLD deepening and the vertical eddy structure, we plot on

the same temporal axis the evolution of the MLD and the vertical temperature gradients inside the eddy core. In the following example we show in Fig.4 the temporal evolution at depth of a long-lived Eratosthenes anticyclone from 2007 to 2009. This kind of anticyclone, also called "Cyprus eddy" or even "Shikmonah gyre", are large mesoscale structures with almost station-ary position south of Cyprus island in the Levantine basin, extensively studied with several CTDs (Brenner, 1993; Moutin and Prieur, 2012), gliders and Argo floats deployments (Hayes et al., 2011). The anticyclonic density anomaly is characterized on

average by a deep (about 400m) and extremely warm temperature anomaly (up to +2°C at 400m) (Moutin and Prieur, 2012; Barboni et al., 2021), sometimes with a strong salt anomaly (Hayes et al., 2011). Thus temperature profiles are considered as a good estimate for relative density and temperature gradient for stratification.

An Argo float remained trapped inside this anticyclone from mid-2008 to the death of this eddy in early 2009, allowing to

capture well an MLD deepening event during winter 2008-2009 (listed hereafter and in Table1 with name ERA1). Temporal resolution for the time series and background in Fig.4a and 4b is 5 days. Figure 4a shows the mixed layer depth evolution, the black line showing the background MLD, with the spread ($20^{th}$ and $80^{th}$ percentiles) as dashed black lines, the red dot being the MLD of the profile colocated inside this anticyclone (see section 3.2). Figure 4b shows the temperature gradient inside the anticyclone as a proxy for stratification. In situ profiles are shown at the beginning (Fig.4c around 12 January 2009) and

the end (Fig.4e around 1 March 2009) of the winter deepening, profiles constituting the background are shown as thin gray





**Figure 4.** In depth evolution of a Eratosthenes anticyclone, listed as 'ERA1' in Table1. (a) MLD evolution, with black continuous line for background MLD (and dashed line for associated spread), anticyclonic core MLD closest to eddy center with red dots. (b) Time series of inside-eddy temperature gradient, blue showing homogeneous and red stratified layers. (c) (respectively (e)) shows vertical profiles around 12 January 2009 (1 March 2009) with background profiles in thin gray lines, background mean as thick gray line, inside-eddy profile as red line, a red dot highlighting anticyclonic core MLD and a green bar indicating homogenized layers (temperature gradient below $2.5 \times 10^{-3}$ $°C.m^{-1}$. Horizontal black line reminds background MLD and spread from panel (a). (d) (respectively (f)) shows profile corresponding position on a map with same color code, together with the eddy maximal speed contour (navy blue shape) and eddy footprint (outermost SSH contour, light blue shape). Topographic data are from ETOPO1 (Smith and Sandwell, 1997).

lines and their mean as thick gray line. Median background MLD (and its spread) is reminded by a continuous (respectively dashed) black horizontal line. Anticyclonic core profile is in red, with a dot for its MLD. Green bars on Fig.4c and 4e highlight the homogenized layers, defining as 'homogenized' a layer with temperature gradient constantly below $2.5 \times 10^{-3}$ $°C.m^{-1}$.





Anticyclonic core profiles have indeed a marked temperature anomaly on the order of +2°C at 450m compared to the back-
ground, proving they indeed sample the eddy core. Some profiles with very warm temperature at 400m deep are misleadingly
considered in the background but do not corrupt the mean (thick gray line). Corresponding situation maps (Fig.4d for January
and 4f for March 2009) show profiles positions, background (inside-eddy) profiles shown as black (red) squares on the maps.
Additionally the anticyclone max speed contour (dark blue shape) and eddy footprint (defined as union of "end" contours at ±
2 days, light blue shape) are also shown for each date on the maps.


First on Fig.4c it should be noticed that the eddy subsurface core in January 2009 is constituted by an extremely homoge-
nized layer that could be identified as mode waters (from 100 to 300m on Fig.4c, but also visible from July 2008 on Fig.4b).
MLD is deeper inside the Eratosthenes anticyclone than its environment, however MLD temporal evolution do not coincide
inside- and outside-eddy. On 12 January 2009, the anticyclonic core MLD is 90m deep, while it is around 60m in the back-
ground, revealing a weak eddy-induced MLD anomaly. The deeper homogenized core remaining unmixed below a seasonal
thermocline : +1 °C temperature jump at 100m on Fig.4c. But the winter cooling and subsequent MLD deepening eroded
this stratification, as shown by the temperature gradient vanishing in the upper 100m (Fig.4b), the winter MLD connects with
the primitive core and mix it. On 1 March 2009 (Fig.4e) the anticyclonic core MLD reached 350m and stays homogeneous
without restratification, whereas in the background other profiles started to restratify and MLD rose back to 40m. Despite noise
in the background spread, one can see that most background profiles (thin gray lines in Fig.4e) are indeed restratified with
temperature gradient in the upper 100m. Anticyclonic core MLD rose back to about 40m only in late March 2009 (see Fig.4a),
revealing a time delay between the start of restratification inside and outside the eddy. It can also be noticed that while the eddy
core remained warm at depth, its mixed vertical profile leads to a negative temperature (and hence positive density) anomaly
from 50m to the surface, compared to the stratified outside profile. Such positive density anomaly above the eddy core forms a
subsurface eddy (Assassi et al., 2016).

Over the whole 2008-2009 winter, background MLD barely reached 60m whereas the anticyclonic core MLD went down to
350m. This intense deepening at the anticyclone core is due to the preexisting subsurface eddy, made of a well mixed layer of
few hundred meter depth below the summer stratification. When the winter MLD is formed, the surface reconnects to this deep
subsurface core and leads to a rapid and strong increase of the MLD in comparison to the eddy background. This configuration
will be called in what follows a 'connecting' MLD.

## 4.2 Winter deepening not connecting preexisting subsurface core

On the other hand, for some other well-sampled anticyclones, it can clearly be seen that the subsurface temperature anomaly
does not connect with the winter deepening, and remains at depth. Figure 5 shows the evolution of another Eratosthenes anti-
cyclone living from 2009 to early 2012, with two recorded anticyclonic core MLD deepening in 2010 and 2012 (respectively
listed in Table1 as 'ERA2' and 'ERA3'), with same color codes than Fig.4, with profiles at 20 March 2010 and 15 June 2010.





**Figure 5.** Same color codes and legend as in Fig.4. In panel (a),(c) and (e), orange lines and dots show other inside-eddy profiles and corresponding MLD, but further away from eddy center than the one shown in red.

As several profiles were located at the same time inside the anticyclone, they are shown in orange line on Fig.5c and 5e (respectively orange dot for MLD on Fig.5a). The red line highlights only the profile with closest distance to the eddy center, assumed to be more representative of the eddy core.

Similarly to the 'ERA1' event in 2009 described above, a thick and deep subsurface anomaly forms primitive eddy core in late 2009, visible on Fig.5b and 5c as an homogeneous layer from 250 to 400m deep ( green bar on Fig. 5c), reaching an anomaly about +2.5°C at 400m . However the anticyclonic core MLD did not deepened in the winter 2009-2010 below 150m, only forming a second homogeneous layer above. This constitutes a second surface core, still separated from the primitive core by a temperature stratification, revealed on Fig.4b by a temperature gradient continuous in time. On the vertical profile on 20





March 2010 (Fig.4) about 1°C temperature jump remains between the two core, forming a double-core anticyclone. In June 2010, this second homogeneous layer is itself covered by the spring 2010 restratification, then forming what is also referred to as 'thermostad', or 'mode-water eddy' in the literature (Dugan et al., 1982). Thanks to the trapped Argo floats remaining

near the eddy core for months, both cores can be tracked until August 2010 as separated in subsurface. This MLD deepening inside anticyclone reveals the possibility of a persisting separation between a primitive subsurface anticyclone core and the new homogeneous layer formed by the current winter mixing, then constituting a double-core anticyclone. Such events are called 'non-connecting' events, and consequences are discussed in Sect.5.2 hereafter, with another clear example in Fig.10.

**4.3   Inside-anticyclone MLD statistics**

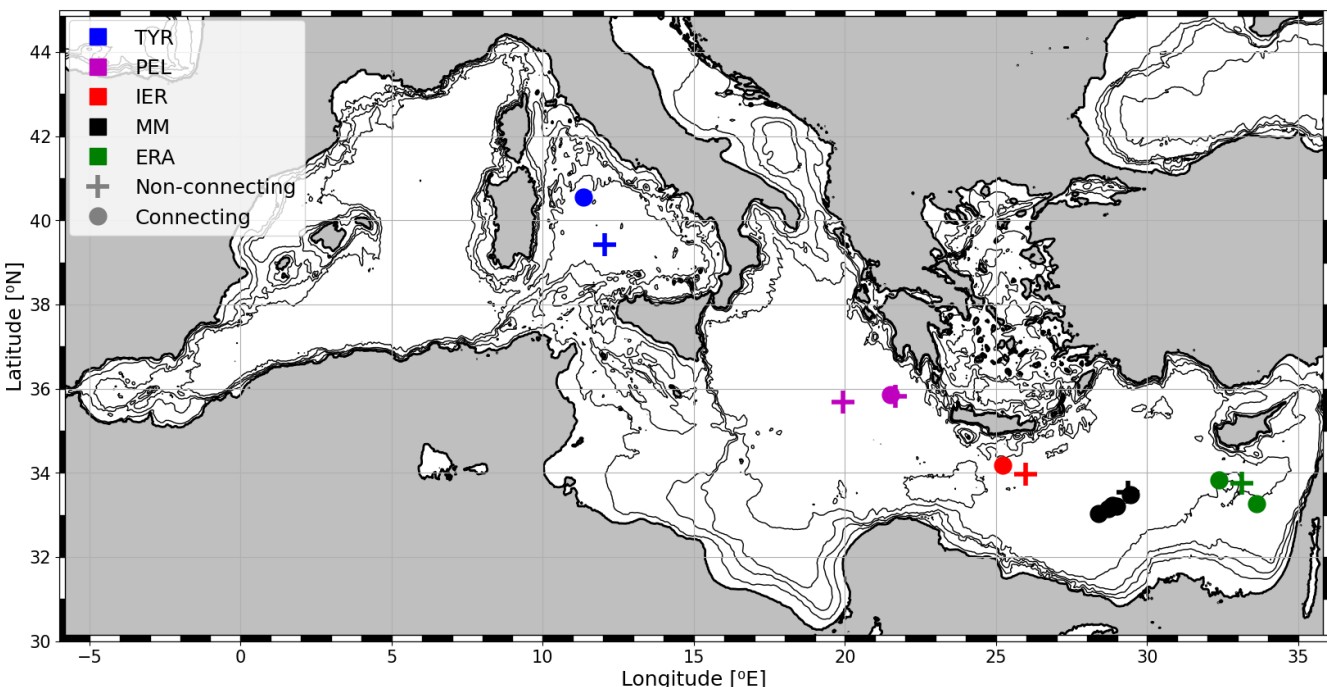

**Figure 6.** Map of well-sampled winter mixed layer deepening events inside anticyclones, with corresponding codes to Table1. Big dots show 'connecting' events, while crosses show 'non-connecting' ones. Color depends on the region : Central Tyrrhenian (TYR), Pelops (PEL), Ierapetra (IER), Mersa-Matruh (MM) and Eratosthenes (ERA, also called "Cyprus"). Isobaths shown on the maps are at 100, 500, 1000 and 1500m deep, topographic data from ETOPO1 (Smith and Sandwell, 1997).

From 2000 to 2021, thanks to extensive Argo deployments sampling eddies, 16 winter MLD deepening events were accurately recorded with vertical profiles over 13 mesoscale eddies, 10 being 'connecting' events, 6 'non-connecting' ones. Several structures were indeed surveyed over 2 winters (see Fig.5 and 10). For each event, the fitting method detailed in Sect.3.3 is




**Table 1.** Main characteristics of the 16 anticyclonic core mixed layer deepening events studied, fitting method and uncertainties are detailed in Sect.3.3. Eddy ID refers to track number in the DYNED Atlas. Note that sometimes two different winters are recorded in the same anticyclone (for example 'IER 1-2'). Event types are 'C' for 'connecting' and 'N' for 'non-connecting'. Year '2018' corresponds to winter 2017-2018. Regions codes, ordered from West to East, stand for : Central Tyrrhenian (TYR), Pelops (PEL), Ierapetra (IER), Mersa-Matruh (MM) and Eratosthenes (ERA). $\Delta MLD$, $t_{max}^{AE}$ and $\Delta\tau$ are illustrated on Fig.3.

| Event | Eddy ID | Type | Year | Position (°N ; °E) | $\Delta MLD$ (m) | $t_{max}^{AE}$ (days since 1 January) | $\Delta\tau$ (days) | Amplitude (cm) | $R_{max}$ (km) | $V_{max}$ ($m.s^{-1}$) |
|---|---|---|---|---|---|---|---|---|---|---|
| TYR1 | 11780 | C | 2018 | 40.6 ; 11.3 | 255± 15 | 50± 3 | 17± 4 | 5.5±0.9 | 38.3±4.3 | 0.24±0.02 |
| TYR2 | 12976 | N | 2020 | 39.4 ; 12.0 | 49± 4 | 28± 4 | 4± 5 | 7.2±3.4 | 33.4±2.6 | 0.18±0.02 |
| PEL1 | 8886 | N | 2015 | 35.7 ; 19.9 | 196± 7 | 79± 4 | 62± 5 | 4.2±0.9 | 42.0±8.7 | 0.19±0.04 |
| PEL2 | 10054 | N | 2016 | 35.8 ; 21.7 | 127± 13 | 61± 10 | 26± 11 | 6.3±1.7 | 38.1±8.7 | 0.31±0.07 |
| PEL3 | 11649 | C | 2019 | 35.9 ; 21.5 | 79± 23 | NaN | NaN | 7.8±0.9 | 39.3±6.6 | 0.36±0.04 |
| IER1 | 11099 | N | 2017 | 34.0 ; 26.0 | 175± 41 | 67± 20 | 46± 20 | 7.7±1.1 | 37.3±4.3 | 0.41±0.05 |
| IER2 | 11099 | C | 2018 | 34.2 ; 25.2 | 211± 12 | 51± 4 | 16± 6 | 7.4±1.6 | 40.9±6.9 | 0.35±0.07 |
| MM1 | 3556 | C | 2006 | 33.1 ; 28.7 | 197± 30 | 40± 20 | 13± 20 | 9.3±1.2 | 45.4±7.5 | 0.41±0.06 |
| MM2 | 4125 | C | 2008 | 33.5 ; 29.4 | 325± 12 | 47± 2 | 36± 8 | 3.7±0.7 | 37.7±5.7 | 0.20±0.04 |
| MM3 | 7656 | C | 2015 | 33.2 ; 28.8 | 236± 7 | 38± 2 | 42± 5 | 8.6±2.2 | 47.8±9.3 | 0.35±0.04 |
| MM4 | 11544 | C | 2018 | 33.2 ; 29.0 | 187± 30 | NaN | NaN | 11.4±1.6 | 61.0±10.0 | 0.38±0.04 |
| MM5 | 11544 | C | 2019 | 33.0 ; 28.4 | 215± 30 | 29± 20 | 13± 20 | 8.7±0.9 | 43.0±4.6 | 0.41±0.05 |
| MM6 | 14400 | N | 2021 | 33.5 ; 29.4 | 151± 12 | 80± 5 | 74± 7 | 3.1±1.1 | 35.9±7.0 | 0.18±0.03 |
| ERA1 | 4914 | C | 2009 | 33.8 ; 32.4 | 338± 30 | 62± 5 | 39± 7 | 2.1±1.0 | 34.9±7.8 | 0.12±0.04 |
| ERA2 | 5906 | N | 2010 | 33.8 ; 33.1 | 136± 7 | 76± 4 | 67± 8 | 3.2±2.0 | 40.1±13.6 | 0.16±0.08 |
| ERA3 | 5906 | C | 2012 | 33.3 ; 33.6 | 180± 30 | 37± 20 | 13± 20 | 4.6±1.0 | 41.8±5.8 | 0.22±0.04 |

applied and parameters are reported in Table1 together with eddy characteristics : eddy SSH amplitude, maximal speed $V_{max}$
and maximal speed radius $R_{max}$. Eddy measurements are estimated by the mean from November to March of the corresponding winter. Figure 6 shows the location of each structure, which actually corresponds to a type of long-lived structures already identified in the literature (Millot and Taupier-Letage, 2005; Hamad et al., 2006; Budillon et al., 2009; Barboni et al., 2021), from West to East : Central Tyrrhenian anticyclone (abbreviated TYR), Pelops (PEL), Ierapetra (IER), Mersa-Matruh (MM) and Eratosthenes (ERA). Position is computed as the mean position during the corresponding winter, even though eddies do not drift a lot in the Mediterranean sea. Crosses on the map shows 'connecting' events, while big dots shows 'non-connecting' ones. Despite regional differences and limited data availability, both types can occur in each region, and provide an observation database allowing statistical comparison. Maximal inside-eddy time $t_{max}^{AE}$ and hence $\Delta\tau$ could not be computed for events MM4 and PEL3, as gaps in the time series do not allow to accurately measure them. However $\Delta MLD$ could always




be computed as in worst cases there are still inside-eddy profiles later in the year allowing to check that maximal MLD was
indeed reached (in similar way than Moutin and Prieur (2012) for previous winter MLD retrieved in April). Both types of
events entail interaction (or not) with a deep surface homogeneous layer (temperature gradient $2.5 \times 10^{-3} \, °C.m^{-1}$), either
a preexisting one (see Fig.4c, 4c and 10c) or a new one (see Fig.5e and 10d). In all winter deepening events listed on Table
1, such homogeneous layers of least 50m were indeed visible on vertical profiles. For Tyrrhenian sea anticyclones (TYR1 &
2) with stronger salinity influence, homogeneous layers with density gradient below $5.0 \times 10^{-4} \, kg.m^{-4}$ are also visible. One
can also notice that 'non-connecting' events are quite common, but double-core structures should be even more frequent, as a
'connecting' event can occur inside a double-core structure and reconnect only the homogeneous core formed in the previous
winter but not the deepest anomaly, as shown later in Fig.10b-e. In other worlds the proportion of 'non-connecting' events is
a lower bound for double-core structures. Map in Fig.6 then reveals the very high occurrence of double-core and mode-water
eddies.


Hausmann et al. (2017) and Gaube et al. (2019) proposed a linear relation between the anticyclonic core MLD anomaly
and its SSH amplitude, using regional average and monthly climatology. We previously showed that MLD anomaly varying
over very short timescales can produce sharp MLD gradient and anomalies reaching several hundreds meters, not captured by
smoothed composites. The relation between MLD anomaly and eddy amplitude is tested on Fig.7a, distinguishing 'connect-
ing' (red dots) and 'non-connecting' (green dots) events, together with Gaube et al. (2019) relation in dashed line ($1m$ MLD
anomaly for $1cm$ eddy amplitude). This proposed relation is obviously not verified, the deep MLD observed in Mediterranean
anticyclones exceeding by far the relation. On the opposite, deepest MLD anomalies seem to be observed in the eddies with
weakest SSH signature. Although surprising at first sight, this trend might be explained by the fact that deepest MLD can be
observed when mixed layer abruptly connects to an anticyclone deep homogeneous core, in subsurface and hidden by a strong
seasonal thermocline. This is in particular the case for 'ERA1' shown in Fig.4, having an extreme $\Delta MLD$ deeper than 300m
but the lowest SSH signature in Table 1.

The relation between the MLD anomaly and the eddy Rossby and Burger numbers are also tested in Fig.7b and 7c. Rossby
number, defined as $Ro = V_{max}/fR_{max}$ where $f$ is the Coriolis frequency, is a non-dimensional measurement of the eddy
intensity. The Burger number, defined as $Bu = (R_d/R_{max})^2$ with $R_d$ the deformation radius (8 to 12 km in the Mediterranean
sea) is a non-dimensional eddy size. Similarly to eddy SSH amplitude, no clear relation can be retrieved, deep and shallow
MLD anomalies appear for various eddy intensity and size and for both 'connecting' and 'non-connecting' events. One can
only notice that 'connecting' events pull MLD deeper in general, and that these events are slightly more observed in large
structures (small $Bu$). Remote-sensing measurements are then hard to link with observed eddy-induced MLD anomalies. On
the opposite the diversity of vertical structure shown in this study (Fig.4, 5 and 10) suggests that eddy vertical structure might
have more influence, and previously proposed linear relation seem to apply mostly for surface-intensified structures.





**Figure 7.** Relationship between maximal MLD anomaly ($\Delta MLD$) and eddy parameters possibly measured through remote sensing : (a) eddy SSH amplitude, compared with proposed $1m$ MLD for $1cm$ SSH relation (Gaube et al., 2019), (b) Rossby number (eddy intensity), and (c) Burger number (non-dimensional eddy size).

## 4.4 Inside-anticyclone restratification delay

A new and important observation is that MLD inside anticyclones tends on average to clearly restratify later than the neighbour-
ing background. It was shown for two individual events in Fig.4 (ERA1, 'connecting') and Fig.5 (ERA2,'non-connecting'), but
it is statistically robust in Table1 : average $t_{max}^{back}$ is 22 days and average $t_{max}^{AE}$ is 49 days, meaning restratification usually begins
in the second half of February in anticyclones, on average one month later than outside-eddy. Restratification delay $\Delta\tau$ can
reach two months in some case : 67 days for ERA2 or 74 days for MM6 (see Fig.9b). Figure 8a shows the relation between $\Delta\tau$





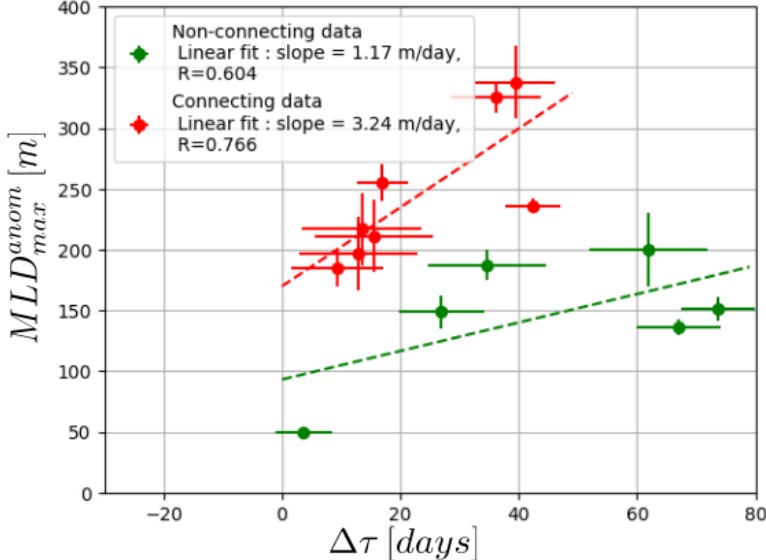

**Figure 8.** Scaling between the maximal MLD anomaly ($\Delta MLD$) and restratification delay $\Delta\tau$ (see scheme on Fig.3), distinguishing 'connecting' (red) and 'non-connecting' (green) events. Linear fit is applied separately and correlation coefficient put in legend. Data and uncertainty are from Table1.

and $\Delta MLD$ and reveals that no clear trend can be identified alone : deep MLD anomalies are observed when the anticyclone

MLD restratify early (low $\Delta\tau$) or later (large $\Delta\tau$). However when distinguishing 'connecting' and 'non-connecting' events, a linear trend appears separately : MLD anomalies go deeper as $\Delta\tau$ increases and for similar $\Delta\tau$ value 'connecting' events go deeper. Linear fit is performed separately for both types, shown by dashed line on Fig.8 : for each day of continued MLD deepening inside anticyclones, a 'connecting' (respectively 'non-connecting') MLD gets about 3m deeper with correlation coefficient of 0.766 (1m deeper with correlation 0.604). This trend is logical, as a later restratification (large $\Delta\tau$) lets the MLD

deepens longer and hence leads to larger $\Delta MLD$.

In order to analyze the MLD evolution together with the mixed layer cooling, Fig.9 shows in the upper panel the MLD fit (see Section 3.3) and in the lower panel the corresponding mixed layer temperature for the PEL2 and MM6 events. Both events are representative of the observed evolution where the temperature can be followed over the whole winter. Maximal background

MLD is reached for PEL2 (respectively MM6) around 30 January 2016 (10 January 2021), marking the end of background mixed layer cooling with a plateau temperature of about 16 °C maintained for about 1.5 months (about 16.5 °C for about 3 months) until restratification. In early 2016 (2021), the anticyclone core is indeed about +1°C warmer than its background, and continues to cool for a while. The inside-eddy maximal MLD is reached around 1 March 2016 (20 March 2021) or +60 (+80) days on Fig.9c (d). Few weeks after, both background and anticyclonic core mixed layers start to warm again around 1

April (in both PEL2 and MM6), but background MLD started to restratify 1 month earlier in PEL2 (2 months earlier in MM6).







**Figure 9.** MLD data and fit , inside- and outside-eddy, illustrated for event PEL2 (a) and MM6 (b) (see Table1). In the lower panel corresponding mixed layer temperature evolution for PEL2 (c) and MM6 (d). A black (red) dashed line marks the time of maximal background (inside-eddy) MLD.

Although it is hard to infer a mechanism from few observations, it seems that the end of restratification outside-eddy does not mean the mixed layer is warmed up again and outside-eddy mixed layer remains cold. The restratification delay seems to be the consequence of a maintained cooling of the initially warmer anticyclone core. Summer heating seems on the other hand to begin at the same time inside- and outside-eddy. Possible mechanism driving this sustained mixing at the anticyclone core are discussed later in section 5.3. An important observation is also that temperature difference between anticyclone core and the background is on the order of $+1$°C while MLD deepens, but almost vanishes (or even get slightly negative) when the mixed layer warms again. Although sparse, these in situ observation are in total agreement with observed eddy SSTA switch by



Moschos et al. (2022) from winter warm-core anticyclones to predominant cold-core anticyclones with spring restratification in the Mediterranean sea.

## 5   Discussion on physical drivers and perspectives

### 5.1   MLD anomaly scaling

We clearly identified the distinction between 'connecting' and 'non-connecting' events as a more important driver than other eddy parameters such as eddy amplitude, surface intensity or size (see Fig.7-8), and this might explain the difficulty to find a general law for any eddy-induced MLD anomaly. Indeed 'connecting' deepening mixed layers seem limited by the bottom of the preexisting subsurface anomaly to which they connect (example in Fig.4e), whereas 'non-connecting' ones by definition do not go enough deep and are then expected to be limited by heat loss, likely also influenced by the eddy. The other important parameter is the restratification delay ($\Delta\tau$), measuring how long the anticyclone continues to deepen MLD or not, and which eventually scale with the maximal MLD anomaly. In a remote sensing perspective, both parameters seem very hard to assess without in situ profiles inside the eddies. Examples shown in this study ( Fig.4, 5 and 10 below) showed the complexity induced by possible connection with previous subsurface anomaly and more generally the key role of the anticyclone vertical structure, that was totally smoothed in previous composite studies. Relationship between eddy-induced MLD anomaly and satellite measurements are definitely more complex. However as theorised by Assassi et al. (2016), detecting remotely information on the eddy vertical structure could be possible, in particular distinguishing the subsurface or surface-intensified nature comparing eddy signature in SSH and SST. ERA1 event in Fig.4 is an almost textbook case of a subsurface eddy with isopycnals doming leading to a cold eddy-SSTA.

### 5.2   Double-core eddy formation

High occurrence of 'non-connecting' events (crosses on Fig.6) are very interesting as they show the formation of double-core anticyclones through winter deepening of the surface layer above a preexisting density anomaly. Double-core eddies were often surveyed in the world ocean (Lilly et al., 2003; Belkin et al., 2020), including in the Western Mediterranean sea (Garreau et al., 2018). Despite various propositions (see e.g. Belkin et al. (2020) for a list ), no clear formation mechanisms emerged. Several studies focused on the so-called 'vertical alignment' of two eddies with different densities in experimental works (Nof and Dewar, 1994), observations (Lilly et al., 2003) or modelling (Trodahl et al., 2020). Interestingly Lilly et al. (2003) observed well in the Labrador sea that double-core anticyclones mostly consist of convective lenses formed in different winters, the heat flux interannual variability leading to different density anomalies, but they explained it with eddies formed separately which later aligned. There were nonetheless previous observations of a second lighter core generated above a preexisting anticyclone. Thanks to repeated XBT transects, Nilsson and Cresswell (1980) surveyed such phenomenon in an anticyclone detached from the East Australian Current, caused by winter heat loss. Bosse et al. (2019) surveyed this in the Lofoten eddy with winter convection, but through glider sections spaced in time, then with a temporal resolution on the order of the month. More recently



Meunier et al. (2018) explained the formation of a double-core Loop Current Eddy by winter diabatic processes. However this
case is different from the Mediterranean anticyclones ; as the Loop Current Eddy consists of an advection of a large structure
of Caribbean waters into the Gulf of Mexico experiencing different surface fluxes with more heat loss and precipitation than
the area where they originate. These diabatic processes by surface winter mixing result in a fresher shallower core above a
saline core of subtropical under waters. Moreover Meunier et al. (2018) explained quantitatively the observed anomaly against
regional average of atmospheric fluxes, whereas in our study the differential MLD evolution between the eddy core and the
background (Fig.4a and Fig.5a) suggested fluxes variations at the scale of the eddy.

What drives the formation of double-core structures should be further investigated, but one could expect the interannual
variability of heat fluxes to be the main driver. This was already suspected by Lilly et al. (2003) (although for them it was for
separate eddies) and Moutin and Prieur (2012). A winter with strong heat loss is expected to deepen MLD a lot, including
inside-eddy, and a subsequently warmer winter could then not achieve to deepen the MLD as much. This mechanism drives
mode-water formation, and it was already shown in other regions, mostly the Atlantic Ocean, that eddies could modulate
mode-water formation (Dugan et al., 1982; Chen et al., 2022). Such hypothesis could also explain the high occurrence of
'non-connecting' events in the Mediterranean sea, this region being known for a high interannual variability of winter heat
loss. Pettenuzzo et al. (2010) indeed found maximal winter heat loss to vary by 20% to 30% (in regional monthly average),
and a plausible connection with the Northern Atlantic Oscillation (NAO). This interannual variability of the heat flux was
already shown to influence deep convection in the North-Western Mediterranean sea (L'Hévéder et al., 2013), and it could then
be expected a higher occurrence of double-core anticyclones due to a stronger Mediterranean sea stratification in a warming
climate (Somot et al., 2006).

An important consequence in the formation of this lighter core in a 'non-connecting' winter deepening is that the second
core is separated from the surface by a thinner seasonal stratification. The next winter is then likely to connect again the new
mixed layer with the upper core, while possibly keeping the primitive deeper core untouched. Such kind of interaction from one
winter to another was observed in the Ierapetra eddy and is presented in Fig.10 (with same color code as in Fig.4a-f and 5a-f).
The Ierapetra eddy a recurrent long-lived and intense anticyclone formed South-East of Crete Amitai et al. (2010); Ioannou
et al. (2017) and recently surveyed by the PERLE campaigns (Ioannou et al., 2019; Durrieu de Madron and Conan, 2019).
Similarly to Eratosthenes anticyclones previously shown, the density anomaly is mostly driven by a warm core, allowing to use
temperature profile as a proxy for stratification (Ioannou et al., 2017). Figure 10 shows the Ierapetra anticyclone formed in au-
tumn 2016. The first winter 2016-2017 turned out to be a 'non-connecting' event (listed in Table1 as 'IER1'). Indeed in March
2017 a preexisting subsurface homogenized layer remains between 350 and 450m anomaly, below the maximal anticyclonic
core MLD of 220m (Fig.10c) and with about $+2$ °C temperature anomaly. From April to December 2017, summer heating
restratified the upper layer and let below a second homogeneous layer between $\sim 100$ and 200m deep, being separated from the
deeper core by a temperature gradient throughout the winter (Fig.10b). In January 2018 (Fig.10d), inside-eddy vertical profile
shows the current winter MLD deepening at 120m - already deeper than the background MLD - and the double-core structure is





still retrieved. The previous winter core still visible between 120 and 200m, albeit being less homogeneous than in March 2017,
and the deeper core from 2016 remains below 350m. At last at the end of February 2018 (Fig.10e), the MLD completely eroded
the seasonal stratification and connects the current MLD with previous winter subsurface core, then reaching about 280m. The
winter 2017-2018 is then a 'connecting' event (listed as 'IER2' in Table1). The timeseries is interrupted inside the anticyclone,
but Argo floats are again colocalized in May 2018, and despite some variability a temperature gradient continuously separates
the two cores between 200 and 250m (see Fig.10b). IER2 was then a 'connecting' event on a double-core structure, but the
deepest anticyclonic core was not mixed. These data bring to light a possible formation process of a double-core anticyclone
through winter convection, and also documents for the first time the fate of the formed subsurface anomaly, which can be
tracked up to the next winter when it was mixed again.

## 5.3 Physical drivers

The observed importance of restratification delay $\Delta\tau$ should also have underlying physical mechanisms. Prolonged MLD
deepening and cooling inside-eddy (see examples in Fig.9) leads to the extreme MLD anomalies sometimes larger than 300m,
and hence marked MLD gradient occurring at scales of the eddy radius (as argued by Gaube et al. (2019), however this is a
composite vision, and one could argue that MLD gradient could occur on shorter scales). Such marked MLD gradients should
trigger mixed layer instabilities leading to restratification (Boccaletti et al., 2007; Fox-Kemper et al., 2008), calling mechanisms
sustaining the mixing inside-eddy during the restratification delay. It should be noted that a homogenized layer itself does not
proof an active mixing, but still reveal the absence of restratification. We also interestingly noticed that in several cases Argo
floats remained well in the anticyclone core during the MLD deepening phase, but often leave the eddy soon after, maybe a
signature for mixed layer instabilities impacting the eddy. The first mechanism explaining longer mixing in anticyclones would
be an eddy modulation of air-sea fluxes by eddy-induced SSTA. Villas Bôas et al. (2015) observed such eddy-modulation on
air-sea sensible and latent heat fluxes, but in regions of energetic surface-intensified eddies, with very warm anticyclones (in
particular the Algulhas current retroflexion). For subsurface anticyclone, the eddy-induced SSTA is on the opposite expected
to be weakened ( see the example of the cold-core anticyclone shown in Fig.4e and the study of Assassi et al. (2016)), and this
mechanism might then not be the most important. MLD deepening enhanced in anticyclones could be explained by other eddy
retroactions than on the heat fluxes, a possible mechanism being the eddy-induced Ekman pumping (Stern, 1965; Gaube et al.,
2015) or enhanced mixing in anticyclones due to near-inertial waves trapping (Kunze, 1985).

## 5.4 Impact on eddy dynamics

Connecting events also raise interesting questions on the consequence of such mixing of deeper subsurface anticyclone core,
in particular the role of inside-eddy convection on the eddy dynamics itself. Studies in the literature mostly focused on winter
convection inside cyclone because of the preconditioning with isopycnals "hinning at their center (Legg et al., 1998; Legg and
McWilliams, 2001). Such phenomenon should also applied for subsurface anticyclone due to the surface isopycnals doming and




thinning (Assassi et al., 2016). The coincidence of observed multiple 'connecting' winters in long-lived anticyclones like the Mersa-Matruh and Eratosthenes structures suggests a possible mechanism regenerating these structure, and maybe explaining the extremely marked cyclone-anticyclone lifetime asymmetry in the Levantine basin (Mkhinini et al., 2014; Barboni et al.,
2021). Interestingly Brenner (1993) already proposed winter cooling as possible mechanism explaining the sustained lifetime of the anticyclone surveyed south of Cyprus. The other structure calling for comparison is the Lofoten eddy in the Sea of Norway, and the Rockwall Trough eddy offshore Ireland, two long-lived deep anticyclonic structures. Winter convection was observed inside the core of the Lofoten eddy, and once thought to help regenerating the structure (Ivanov and Korablev, 1995; Köhl, 2007; Bosse et al., 2019). Double core formation was also observed in the Lofoten eddy (Bosse et al., 2019). Recent numerical
studies showed this regeneration was primarily driven by merging of smaller structures (Köhl, 2007; Trodahl et al., 2020), however de Marez et al. (2021) showed that winter time convection eased this merging process by deepening of the eddy core. Merging of eddies detached from the coast towards an offshore anticyclonic attractor being also observed in the Levantine basin (Barboni et al., 2021), this could provide another explanation to long-lived Mediterranean anticyclones. Cylone-anticyclone asymmetry might not have just one mechanism, as other arguments were already proposed. Anticyclones have indeed a larger
radius and are more coherent.

## 5.5 Biological impacts inside anticyclones

Double-core anticyclone formation ('non-connecting' events) letting a deep homogeneous layer unmixed could have key biological consequence. Following Moutin and Prieur (2012), it keeps the surface euphotic layer away from the deep nutriment source, the main nitracline staying at the eddy bottom, and in double-core eddies a secondary nitracline should form at the
505 top of the deeper core. The whole system would then evolve towards an ultra-oligotrophic system because of nutrients being very weakly injected to the euphotic layer. This is expected to be in particular the case when a primitive subsurface core is not connected to the surface for several winter, such as the example of the Ierapetra anticyclone in Fig.10. Anticyclone ventilated in winter were already observed to store enhanced dissolved organic carbon at depth (Krom et al., 1992; Moutin and Prieur, 2012). Krom et al. (1992) estimated an annual nutrient budget for the eddy (which is an Eratosthenes anticyclone sampled
in 1989) but as already discussed by Moutin and Prieur (2012) this will highly depends on the preexisting vertical structure and on whether the main nitracline is reached. On the other hand this aphotic layer remaining unconnected from the surface could provide a pathway for dissolved organic carbon at depth. Such forecast evolution towards an ultra-oligotrophic system in Mediterranean anticyclones would depend on the ratio of 'connecting' versus 'non-connecting' events, and as previously discussed in section 5.2, this trend could be fostered in a warming Mediterranean sea.

## 6 Conclusions

In this paper we were able to analyze, thanks to a combination of satellite observations and numerous in-situ data, several time series that finely describe the evolution of the winter mixed layer in the core of Mediterranean anticyclones. We even succeeded in following, for the same long-lived anticyclone, the evolution of its MLD over two consecutive years. This allowed





us to quantify extreme anomalies induced by mesoscale eddies in the mixing layer, which would have been smoothed in a standard composite analysis. Indeed, we observed that the winter mixing layer can go down to 380 m in the core of Levantine Basin anticyclones, while the surrounding background MLD does not go deeper than 80 m or 100m.

We also observed a time lag of several weeks, and sometimes up to two months, in the spring restratification between the core of these deep anticyclones and the background sea, revealing that MLD temporal evolution is not uniform. Indeed, when the later restratifies due to rising temperature of the atmosphere, the core of these mesoscale anticyclones which are warmer continues to deepen and to cool. This time lag induces very strong spatial heterogeneities of the MLD in the eastern Mediterranean Sea during the early spring, with observed maximal MLD ranging from 50 to 330m.

We showed that this localized deepening of the MLD is controlled by the vertical structure of these eddies. When the surface mixing layer connects with the subsurface core of preexisting anticyclones, a rapid deepening of the surface mixed layer is observed. Conversely, when the surface mixed layer does not connect with the subsurface core, a double-core eddy is formed. Connection or not with preexisting subsurface core prove to be more relevant to describe MLD deepening than other eddy parameters such as SSH amplitude or size. MLD anomalies was observed to linearly increase with restratification delay, but increasing roughly 2 to 3 times faster for 'connecting' MLD than 'non-connecting' one.

These extreme MLD deepenings in anticyclone cores reveal complex and rich interaction between surface and subsurface of the eddies. Connection between the mixed layer and subsurface anomalies provide a way to propagate heat at depth while mixing in winter, which consequences remain to investigate. These winter deepening inside anticyclones could also play a role in sustaining the extremely long-lived anticyclone in the Eastern Mediterranean. MLD anomalies in cyclonic eddies remain to be investigated, and an open question would be to know if a restratification delay could also be observed in cyclones.

*Data availability.* CORA DT profiles (Szekely et al., 2019b) are freely available online on Copernicus marine service (CMEMS, https:// marine.copernicus.eu/ ) under product name INSITU_GLO_TS_REP_OBSERVATIONS_013_001_b. Copernicus NRT profiles (Copernicus, 2021) are freely available on CMEMS under product name INSITU_GLO_NRT_OBSERVATIONS_013_030. DYNED Atlas eddy altimetric detections and contours from 2000 to 2019 is available at https://doi.org/10.14768/2019130201.2. AVISO SSHNRT day+6 1/8°data (Pujol, 2021) are freely available on CMEMS under product name SEALEVEL_EUR_PHY_L4_NRT_OBSERVATIONS_008_060 AMEDA eddy tracking algorithm is open source and available at https://github.com/briaclevu/AMEDA.

*Author contributions.* AB and SC built the methodology, performed the data analysis and investigation, wrote the manuscript and contributed equally to this work. AS supervised and validated the study and did funding acquisition. BL processed and produced eddy detections. FD provides in situ data in cooperation with SHOM cruises.



*Competing interests.* All authors declare they have no conflicts of interest.

*Acknowledgements.* The authors gratefully acknowledge the *Délégation Générale de l'Armement* which funded the program Protevs II into which the PERLE campaigns were scheduled, the French Naval Hydrologic and Oceanographic Service (SHOM) and the crew of the RV *L'Atalante* (Ifremer) for their contribution to the PERLE1 campaign, the crew of the RV *Pourquoi Pas ?* for their contribution to the PERLE2 campaign, and the one of the RV *Beautemps-Beaupré* for their contribution to the opportunity Arvor deployments in the Mersa-Matruh eddy
in February 2021. The Argo data were collected and made freely available by the International Argo Program and the national programs that contribute to it (http://argo.jcommops.org/).



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







**Figure 10.** Same color codes and legend as in Fig.4 and 5, but for a Ierapetra anticyclone formed in 2016. Three vertical sections show respectively the mixing in early 2017 not reaching the deep subsurface core (panel c), the winter in early 2018 with a double-core, the shallower from winter 2017-2018 and the deeper still untouched (panel d) and at last the MLD deepening in March 2018 connecting the anomaly from March 2017 with the surface (panel e).





## Appendix A: In situ profile checking methodology

In both DT (Szekely et al., 2019b) and NRT (Copernicus, 2021) datasets vertical profiles data coming from XBT, CTD, glider
and profiling floats are collected by selecting files with respective data type codes **XB, CT, GL, PF**. When a profile from 2000
to 2019 was available in both DT and NRT mode, it was retrieved from the CORA-DT dataset which performs more quality
checks (Szekely et al., 2019a). Selection was done with the following steps, separately for temperature and salinity, and when
available, 'ADJUSTED' properties are collected :

- Position and date quality control (QC) flags equal to 1,2 or 5 and position not on land.

- First valid value (QC=1,2 or 5) above 50m, last valid value below 400m, and 40 measurements between 50 and 400m.

- Temperature data below 12° or above 35° are discarded, salinity data below 30 PSU or above 42 PSU are discarded. These
  parameters are specific to the Mediterranean sea and allow further selection.

- When both temperature and salinity are available, density is computed using the TEOS-10 equation (McDougall et al.,
  2009) from the Python package `gsw` (https://teos-10.github.io/GSW-Python/)

- Profiles are interpolated on the same vertical grid, with 5m grid step from 5m to 300m depth and 10m grid step from
  300m to 2000m. Maximal gap allowed is 20m, and profiles with gaps are discarded.

- Profiles with temperature jumps over 5m higher than +6°C or -2°C are discarded, as assumed to be unrealistic. This is
  required in particular to filter out noisy XBT profiles.

- After these steps, only profiles with more than 40 data on the interpolated grid between 50 and 400m are kept.





**Figure A1.** Sensitivity of the background MLD on the different parameters for events IER1-2 (see Table1 and Fig.10a) : (a) $\Delta day$ (day interval) and (b) $\Delta y$ (year interval). (c) Sensitivity to the MLD computation method. The background MLD method used throughout this study is $\Delta day = 10$ days, $\Delta y = 1$ year and the gradient method (common blue line on panels a-c).