# Peer review of "How subsurface and double-core anticyclones intensify the winter mixed layer deepening in the Mediterranean sea"

_EGUsphere, 2022_

## Referee Comment (RC1)

Barboni et al. 2022, Review by A. Capet

The authors investigate the vertical structures of Mediterranean anticyclones and in particular the temporal evolution of the mixed layer depth in their core, with respect to that of the surrounding background sea state. The major originality of this research compared to similar studies is that the authors could gather sufficient in-situ data (multi-platform) to explore this topic on individual instances, rather than adopting the usual composite framework. Free of the smoothing effect of composite approaches, they are able to demonstrate important dynamics in the anticyclone evolution through the winter deepening of the mixed layer depth. In particular, they highlight the difference between cases where MLD deepening inside eddy cores is strong enough to reconnect with pre-existing subsurface homogeneous layer and cases where the subsurface homogeneous layer remains unperturbed, leading to the formation of multi-core anticyclones, with subsurface cores piling up from year to year. In my view, this study provides fresh insight into the process of anticyclone formation and evolution from year to year, as well as new keys for the interpretation of subsurface water masses' history. The manuscript is well written, although it could certainly be streamlined for more efficient delivery of the key results. I recommend publication, after a careful edition aiming for efficient reading. Some bits of advice in that sense are provided below.

MAJOR
* * *
1. The result section is hard to read as the author is 'very close to its subject', and directly addresses very specific graphical elements of the provided plots without the general, larger perspective statements needed to bring the reader there with him. I would advise some reformulation with those general guidelines in mind: 1) Before entering the details, explain briefly what you're going to show and why. 2) Avoid detailing graphical keys in the text, use the captions for this. 3) Focus the text on the key elements that the readers should acquire to apprehend the discussion.
2. Is Sect 4.1 (as an example) intended to describe one specific event (ERA1) or rather one type of event (connecting pre-existing subsurface core)? In the first case, mention "example of … " in the section title. In the second case, it would be good to end the section with statements about such types of phenomena that are not specific to the chosen example.

MINOR
* * *
- (Sect 2.1): Mention here already the multi-platform nature of the datasets (Argo, XBT, gliders, etc ..). It would then be relevant to provide the relative abundance of profiles obtained from each platform type. Also mention here explicitly, although briefly, that DT profiles are favored in case of spotted duplicates with NRT datasets.
- L122: It may be worth providing this website as a hyperlink.
- It would make sense to switch Sect 2.1 & 2.2, to have MLD extraction methods presented just after the profile dataset from which MLD is extracted.
- L138: Instead of 'this' use explicitly 'the main thermocline' (if I've understood correctly).

- L140: Please state explicitly the reasons that led to your choice of method. For instance, 'As we aim to characterize MLD as the lower boundary of the upper mixed layer while accounting for small-scale restratification event, we opted for the following methodology mixing the threshold and gradient approaches.' (rephrase as required).
- L145 "than" -> "then"
- L152-153: The problem is not that there are double-gradient profiles, but rather that the threshold method overestimates MLD in that case. I'd rephrase, for instance: "[..] 22 of them were identified as double gradient profiles, and led to an overestimation of the MLD when using the threshold method.". The next sentence suggests that switching the method reduces the number of double-gradient profiles. Instead, it reduces the number of cases where these situations resulted in an MLD overestimation. Please rephrase this to solve this confusion.
- L164: 'is' -> 'if'
- Eq(2), first line: I would expect to retrieve the form of Eq (1), in the cases where $\tau_1 = \tau_2$, or if B=0, but this isn't the case with the given form. Shouldn't the parentheses (t-tmax) be squared?
- Sect4. : It would ease the reading to include a few preamble lines to Sect 4 presenting the different cases explored, before jumping to the specific subsections.
- L251: Here and elsewhere, avoid providing the graphical keys of figures in the main text. That's what captions are made for.
- L262: Red (as in the text) or orange (as in the figure caption inset)?
- L267: remove "extremely". What is 'mode waters'?
- L266-273: This is very hard to read. Please attempt rephrasing.
- L277: Please provide the delay.
- L279: '[..] forms a subsurface eddy' or "has been used as a criterium to define subsurface eddies"?
- L301 "4b" -> "5b".
- L302 (Fig4)
- L326: "(temperature gradient **<** 2.5…)" ?
- L396: "enough deep" -> "deep enough".
- L403: 'comparing' -> 'by comparing'?
- L444: "The Ierapetra eddy 'IS' a recurrent … " + give refs inside parentheses.
- L469: 'calling' -> "which calls for"
- L485 "thinning"?
- L519/520: I'd used 'mixed layer' instead of 'mixing' layer. As you mentioned earlier, you only have access to density/temperature gradients, not actual mixing rates.
- L537: Not only heat can be exchanged. For instance, I believe this mechanism to be particularly relevant for oxygen studies.
- L538: "with consequences that remain to be investigated".

FIGURES
* * *
- FIG1:
    - MLD_{\sigma} and MLD_{T} should be defined explicitly in the text (e.g. L159-160).
- FIG2 :

- - ○ "One" -> "profile" (2 times)
  - FIG4 :
    - ○ The (very small) legend on the map insets mentions 'profiles inside the anticyclones' in orange. Yet, those appear as black on the figure? Or is it that all the indicated profiles don't meet the "inside" criterion? But then some of those should be 'ambiguous' profiles, no? Please check carefully.
  - Table 1:
    - ○ For MM4, use a dash "-", instead of "NaN".
  - A1 :
    - ○ There seems to be a mistake between a) ($\Delta$ years) and  b) ($\Delta$ days).
    - ○ Adopt the same blue color for the selected parameters and method in each panel.

---

## Referee Comment (RC2)

**Comment on egusphere-2022-649 titled as: How subsurface and double-core anticyclones intensify the winter mixed layer deepening in the Mediterranean Sea" by Alexandre Barboni et al.,**

The temporal evolution of MLD in the Mediterranean Sea is investigated in this ms. It is shown that the MLD restratification delay and connection with preexisting subsurface anomalies appear to be determinant in MLD modulation by mesoscale eddy and highlights the importance of interaction with eddy vertical structure. The study is novel and will advance our understanding of the impact of mesoscale eddies on the dynamics of seawater properties. The manuscript is well structured and discussed in detail. I recommend publication, after minor modification and changes. The general and specific comments are given below.

**Abstract**

In general, the quality of abstract is not keeping with the entire ms and requires couple of modifications. It is recommended to review the abstract and re-written it, specifically form line 1 to 12.

Some few examples are given below:

Line 2: ....., shoaling very: check the grammatic of the sentence, and replace the comma with an and.

Line 2-6, I suggest to focus on your achievements rather than what is done previously (this is more in the introduction section), if it is really important to mention, put it in a way that shows what you have improved in comparison to previous works.

Line 13: cooling of MLD, does not make much sense, do you mean reducing the strength of that?

please be more specific: Line 13-14: how often it is reaching to more than 2 months. Line 10: at a fine temporal scale on the order of week, you could directly mention how many week(s) ?

Line 16: it is not in the opposite of the former sentence, rather an additional information.

**Introduction:**

Line 40-50: He et al., 2018 show that eddy amplitude is related to surface T anomalies with different behavior for AE and CE, how does their finding cooperate to Gaube et al., 2019?

The ms is focused on the subsurface mesoscale eddies, therefore, it is suggested to briefly explain the subsurface eddies and their differences with other typical type of dominant eddies in the Mediterranean Sea.

line 70: I was looking to read the reasons why the MS is an interesting region to study eddy influence on MLD. The text is very scatter and does not clearly explain why MS is an interesting region.

Line 77: remove the sentence here: -All these structures should have a different impact on the mixed layer. You are seeking to find this in the ms.

**Data:**

Are the 157053 profiles unique data or some data are repeated in your data bank?

Line 112: why 2 set of different data sets are used?

Line 110-125: please name couple of successful application of using AMEDA on detecting/analyzing eddies in the other regions apart from MS (and preferably not from the co-author of this ms).

Figure 2: D--->D0

Line 190-195: why did you choose the background of an eddy by time/spatial averaging with the given time and radius, rather than climatological averaging at the location of the eddy by removing the eddy events following previous studies such as Gaube et al., 2019.

**Results and discussions**

Figures 2,4 and 5: It is suggested to change the colorbar specially from 0 to -5000 m, as it may get confused with the eddy contours or alternatively change the eddy contours line colors.

Figure 4-5: Keep the unit in the figures and text consistent, either °C/m (it is suggested as the profile depths does not cover a km) or °C/km. Does the red region quality in figure 4-5 between 0 and 100 m depth improve, if the colorbar covers a larger number for example to -30 instead of -20?

Would it be possible to add an extra column to table 1 with the eddy life time since the generation day?

Please pay more attention in using abbreviations in the entire text. An example of inconsistency: PEL is first introduced in Fig3 and line 231 without explaining! Then in line 318.

Figure 6: would it be possible to add the eddy path trajectory for the indicated eddies in this figure?

The section 5.5 is suggested to be removed from the ms as i) it is not well discussed ii) out of the focus of the ms.

**Appendix:**

How does the quality control algorithm work? How does it remove bad quality data? How to you define bad data/spikes in T/S profiles?

---

## Author Comment (AC1)

**Barboni et al. 2022, Review by A. Capet**

The authors investigate the vertical structures of Mediterranean anticyclones and in particular the temporal evolution of the mixed layer depth in their core, with respect to that of the surrounding background sea state. The major originality of this research compared to similar studies is that the authors could gather sufficient in-situ data (multi-platform) to explore this topic on individual instances, rather than adopting the usual composite framework. Free of the smoothing effect of composite approaches, they are able to demonstrate important dynamics in the anticyclone evolution through the winter deepening of the mixed layer depth. In particular, they highlight the difference between cases where MLD deepening inside eddy cores is strong enough to reconnect with pre-existing subsurface homogeneous layer and cases where the subsurface homogeneous layer remains unperturbed, leading to the formation of multi-core anticyclones, with subsurface cores piling up from year to year. In my view, this study provides fresh insight into the process of anticyclone formation and evolution from year to year, as well as new keys for the interpretation of subsurface water masses' history. The manuscript is well written, although it could certainly be streamlined for more efficient delivery of the key results. I recommend publication, after a careful edition aiming for efficient reading. Some bits of advice in that sense are provided below.

**MAJOR**
* * *
1. The result section is hard to read as the author is 'very close to its subject', and directly addresses very specific graphical elements of the provided plots without the general, larger perspective statements needed to bring the reader there with him. I would advise some reformulation with those general guidelines in mind: 1) Before entering the details, explain briefly what you're going to show and why. 2) Avoid detailing graphical keys in the text, use the captions for this. 3) Focus the text on the key elements that the readers should acquire to apprehend the discussion.
2. Is Sect 4.1 (as an example) intended to describe one specific event (ERA1) or rather one type of event (connecting pre-existing subsurface core)? In the first case, mention "example of … " in the section title. In the second case, it would be good to end the section with statements about such types of phenomena that are not specific to the chosen example.

      We thank the reviewer for these constructive comments and his interest in the submitted study. Results section of the manuscript was rewritten for more readability in this sense, letting more descriptive details to the figures captions (which fonts are slightly increased). In particular an introductory paragraph opens Section 4 before detailing the two winter mixed layer evolution patterns observed ('connecting' and 'non-connecting'), that are illustrated by the examples in Fig.4 (Eratosthenes-1) and Fig.5 (mostly Eratosthenes-2). Transitions between results subsections are also introduced.

**MINOR**
* * *
1. (Sect 2.1): Mention here already the multi-platform nature of the datasets (Argo, XBT, gliders, etc ..). It would then be relevant to provide the relative abundance of profiles

obtained from each platform type. Also mention here explicitly, although briefly, that DT profiles are favored in case of spotted duplicates with NRT datasets.

We were indeed quite brief on the dataset description, as it was intended to be published separately in a data paper, in order to analyze with more details the mean eddy impact on climatology. The following figures should answer this question :

[Figure]

[Figure]

2. L122: It may be worth providing this website as a hyperlink.

3. It would make sense to switch Sect 2.1 & 2.2, to have MLD extraction methods presented just after the profile dataset from which MLD is extracted.

The sections were switched and the paragraph on the description of the sections in the introduction was modified to stay consistent.

4. L138: Instead of 'this' use explicitly 'the main thermocline' (if I've understood correctly).

5. L140: Please state explicitly the reasons that led to your choice of method. For instance, 'As we aim to characterize MLD as the lower boundary of the upper mixed layer while accounting for small-scale restratification event, we opted for the following methodology mixing the threshold and gradient approaches.' (rephrase as required).
   Modified by 'To capture such small-scale restratification events, we built the following methodology combining both threshold and gradient approaches'

6. L145 "than" -> "then"
7. L152-153: The problem is not that there are double-gradient profiles, but rather that the threshold method overestimates MLD in that case. I'd rephrase, for instance: "[..] 22 of them were identified as double gradient profiles, and led to an overestimation of the MLD when using the threshold method.". The next sentence suggests that switching the method reduces the number of double-gradient profiles. Instead, it reduces the number of cases where these situations resulted in an MLD overestimation. Please rephrase this to solve this confusion.

   Rephrase by : "22 (5.5 \%) of them were identified as double gradient profiles, resulting in an overestimated MLD when derived with the threshold method. Moving to our methodology, this issue is now only encountered for 2 profiles (0.5 \%)."

8. L164: 'is' -> 'if'

9. Eq(2), first line: I would expect to retrieve the form of Eq (1), in the cases where \tau_1 = \tau_2, or if B=0, but this isn't the case with the given form. Shouldn't the parentheses (t-tmax) be squared?

   We chose a fitting function for the MLD deepening inside-eddy that is not a double Gaussian, but instead a double exponential (Eq.2). Fitting function for the background MLD (Eq.1) is then not retrieved in the limit B=>0, however we used a double exponential because it allows a degree of freedom on the MLD time derivative. I.e. df/dt is not bounded to 0 in $t=t_{max}^{AE}$

10. Sect4. : It would ease the reading to include a few preamble lines to Sect 4 presenting the different cases explored, before jumping to the specific subsections.

11. L251: Here and elsewhere, avoid providing the graphical keys of figures in the main text. That's what captions are made for.

12. L262: Red (as in the text) or orange (as in the figure caption inset)?

13. L267: remove "extremely". What is 'mode waters'?
    "mode waters" was indeed weakly defined and we preferred to describe this homogenous layer as "likely formed by convection the previous winter".

14. L266-273: This is very hard to read. Please attempt rephrasing.

15. L277: Please provide the delay.

16. L279: '[..] forms a subsurface eddy' or "has been used as a criterium to define subsurface eddies"?

   Indeed such positive density anomaly "has been used as a criterium to define subsurface eddies" (see Assassi et al 2016)

17. L301 "4b" -> "5b".

18. L302 (Fig4)

19. L326: "(temperature gradient < 2.5…)" ?

   Indeed, and to be more precise, temperature gradient below 2.5 x 10^(-3) ⁰C/m in absolute value

20. L396: "enough deep" -> "deep enough".

21. L403: 'comparing' -> 'by comparing' ?

22. L444: "The Ierapetra eddy 'IS' a recurrent … " + give refs inside parentheses.

23. L469: 'calling' -> "which calls for"

24. L485 "thinning"?

   "Thinning" was indeed not accurate, we preferred "isopcycnals doming and subsequent stratification weakening" in the revised manuscript.

25. L519/520: I'd used 'mixed layer' instead of 'mixing' layer. As you mentioned earlier, you only have access to density/temperature gradients, not actual mixing rates.

26. L537: Not only heat can be exchanged. For instance, I believe this mechanism to be particularly relevant for oxygen studies.
   This mechanism could be very important for oxygen, but also nutrient input to the euphotic layer. Discussion in section 5.5 is improved in this sense. We refer in particular to Krom et al (1992) and Moutin and Prieur (2012) who both observed a 'connecting' mixed layer in 1989 and 2008.

27. L538: "with consequences that remain to be investigated".

FIGURES
* * *
1. FIG1: MLD_{\sigma} and MLD_{T} should be defined explicitly in the text (e.g. L159-160).

   Introduced in the text

2. FIG2 : "One" -> "profile" (2 times)
3. FIG4 :
    a. The (very small) legend on the map insets mentions 'profiles inside the anticyclones' in orange. Yet, those appear as black on the figure? Or is it that all the indicated profiles don't meet the "inside" criterion? But then some of those should be 'ambiguous' profiles, no? Please check carefully.

        In situ profiles labeled as 'inside' must indeed be cast inside the structure at the considered date. However profiles constituting the background are outside of any eddies (at ±2 days as explained in Section 3.2) at the same time of the year at ±10 days, but can be 1 year later or earlier. They can then appear inaccurately as inside a structure because they were cast at a time no detected eddies was present.
        Note that profiles positions for the background slightly changed in maps in Fig.4,5 and 10  due to a small code mistake that does not affect the analysis.

    b. Table 1:
        For MM4, use a dash "-", instead of "NaN".
4. A1 :
    ○ There seems to be a mistake between a) (\Delta years) and b) (\Delta days).
    ○ Adopt the same blue color for the selected parameters and method in each panel.

---

## Author Comment (AC2)

**Comment on egusphere-2022-649 titled as: How subsurface and double-core anticyclones intensify the winter mixed layer deepening in the Mediterranean Sea" by Barboni, Coadou-Chaventon et al.,**

The temporal evolution of MLD in the Mediterranean Sea is investigated in this ms. It is shown that the MLD restratification delay and connection with preexisting subsurface anomalies appear to be determinant in MLD modulation by mesoscale eddy and highlights the importance of interaction with eddy vertical structure. The study is novel and will advance our understanding of the impact of mesoscale eddies on the dynamics of seawater properties. The manuscript is well structured and discussed in detail. I recommend publication, after minor modification and changes. The general and specific comments are given below.

**Abstract**

1. In general, the quality of abstract is not keeping with the entire ms and requires couple of modifications. It is recommended to review the abstract and re-written it, specifically from line 1 to 12.
   Some few examples are given below:

2. Line 2: ....., shoaling very: check the grammatic of the sentence, and replace the comma with an and.

3. Line 2-6, I suggest to focus on your achievements rather than what is done previously (this is more in the introduction section), if it is really important to mention, put it in a way that shows what you have improved in comparison to previous works.

4. Line 13: cooling of MLD, does not make much sense, do you mean reducing the strength of that? please be more specific:

5. Line 13-14: how often it is reaching to more than 2 months.

6. Line 10: at a fine temporal scale on the order of week, you could directly mention how many week(s) ?

7. Line 16: it is not in the opposite of the former sentence, rather an additional information.

> The upper comments have been taken into account and the abstract have been re-written in that sense. Particular attention has been put in underlining what are the main results from this study. More quantitative details regarding our results were added (eg. temporal scale of 10 days, 3 cases out of 16 where the MLD restratification delay is more than 2 months)

**Introduction:**

8. Line 40-50: He et al., 2018 show that eddy amplitude is related to surface T anomalies with different behavior for AE and CE, how does their finding cooperate to Gaube et al., 2019?

> We greatly thank the reviewer for getting to our knowledge this interesting article. He et al (*A New Assessment of Mesoscale Eddies in the South China Sea: Surface Features, Three-Dimensional Structures, and Thermohaline Transports*, 2018) showed that

eddy amplitude is related to the temperature **subsurface** anomaly (based on their Fig.16). They still observed slightly warmer SST inside anticyclone, a vision maybe blurred by the eddy stirring in surface, then making the inside-eddy SST signature more similar to a dipole. This result is different to the one of Gaube et al (2019) and especially Haussmann and Czaja (2012), however not contradictory : the composite method provides an average shape and vertical structure that is scaled with the eddy amplitude.

Very interesting is the fact that they indeed found eddy-induced MLD anomalies for anticyclones predominantly in subsurface, but somehow small and did not provide a trend. This limit is probably due to their method of an 'annual' MLD anomaly by removing the MLD seasonal cycle from a monthly climatology.

From our work in the Mediterranean sea, we think that much greater MLD anomalies could be observed in a region with subsurface eddies such as the South China sea attempting a Lagrangian tracking of eddies. In particular one could interestingly notice a higher PDF for anticyclone MLD around 120m in their Fig.11b, that could be some strong but brief winter MLD smoothed in the annual average.

9. The ms is focused on the subsurface mesoscale eddies, therefore, it is suggested to briefly explain the subsurface eddies and their differences with other typical type of dominant eddies in the Mediterranean Sea.

10. line 70: I was looking to read the reasons why the MS is an interesting region to study eddy influence on MLD. The text is very scatter and does not clearly explain why MS is an interesting region.

Pt 9-10 : A clearer definition of subsurface anticyclone was added and following paragraph about the reasons to study Mediterranean anticyclone was rewritten. Two main arguments can be listed :
- A high-density of in situ observation allowing to follow in time inside-eddy evolution for particular structures, and then beyond a composite vision
- Wide variety of eddy dynamical behaviors

11. Line 77: remove the sentence here: -All these structures should have a different impact on the mixed layer. You are seeking to find this in the ms.

**Data:**

12. Are the 157053 profiles unique data or some data are repeated in your data bank?

We added a sentence (in addition to the Appendix) to clarify this : duplicates are checked between CORA-DT and Copernicus-NRT. In case a duplicate is detected, it is retrieved from the delayed-time repository.

There are : 113486 profiles in CORA-DT from 2000 to 2019 ; 20746 profiles in Copernicus-NRT from 2000 to 2019 and 22821 profiles in Copernicus-NRT from 2020 to 2021

13. Line 112: why 2 set of different data sets are used?

   Delayed-time dataset (CORA-DT) is supposed to be more reliable with more quality controls. However if delayed-time profiles are supposed to be published 6 months after the cast date, some data are often released 1 or 2 years later. Collecting profiles from the second dataset Copernicus-NRT allows accurate data sampling in 2020 and 2021, enabling to measure for instance the Mersa-Matruh-6 event.

   Additionally, we realized that CORA-DT quality controls removed profiles with bad salinity data (QC flag '4') even though the temperature is good and exploitable. This occurred for instance for Argo float 6903204 released in the Ierapetra anticyclone in September 2017. Winter deepening event Ierapetra-2 would then not have been measured retrieving data from CORA-DT only.

14. Line 110-125: please name couple of successful application of using AMEDA on detecting/analyzing eddies in the other regions apart from MS (and preferably not from the co-author of this ms).

   AMEDA is an open source eddy tracking algorithm (https://github.com/briaclevu/AMEDA) . It has been successfully applied in both altimetric data and numerical simulation outputs, and in various regions, such as the Arabian sea (de Marez et al. 2019), the Mediterranean sea (Barboni et al. 2021) or the Northern-Eastern Atlantic Ocean (de Marez et al. 2021 : The influence of merger and convection on an anticyclonic eddy trapped in a bowl).

   The main different with other commonly used eddy detection algorithms (Chelton et al (2008), Chaigneau et al (2009), Mason et al (2014)) is that AMEDA does not consider sea surface height but velocity contours. If a geostrophic approximation is considered, velocity contours are equivalent to SSH isolines, but AMEDA is then applicable in other experiments, such as particle image velocimetry (PIV).

15. Figure 2: D--->D0
16. Line 190-195: why did you choose the background of an eddy by time/spatial averaging with the given time and radius, rather than climatological averaging at the location of the eddy by removing the eddy events following previous studies such as Gaube et al., 2019.

   We didn't built the background with climatological averaging in order to avoid as much as possible the time-averaging and keep the interannual variability. Since we compare the temporal evolution of MLD within eddies with respect to that of the surrounding background sea state, a smoothing of the background state will result in an erroneous MLD modulation by eddies estimate, in particular for years where the interannual variability is strong

**Results and discussions**

17. Figures 2,4 and 5: It is suggested to change the colorbar specially from 0 to -5000 m, as it may get confused with the eddy contours or alternatively change the eddy contours line colors.

18. Figure 4-5: Keep the unit in the figures and text consistent, either °C/m (it is suggested as the profile depths does not cover a km) or °C/km.

19. Does the red region quality in figure 4-5 between 0 and 100 m depth improve, if the colorbar covers a larger number for example to - 30 instead of -20?

> Pt 17 to 19 : Figures were improved in this sense. A mistake was also corrected in Fig.5e, with accurate profiles in April 2010. Position of background profiles also slightly changed because of a code mistake in background selection parameters, but this does not alter the analysis.

20. Would it be possible to add an extra column to table 1 with the eddy lifetime since the generation day ?

> The corresponding column was added in the revised manuscript. The shortest one lived for 345 days, the oldest for 1229. Please also note that eddy tracking for Mersa-Matruh-6 stopped only because the dataset ends in December 2021

21. Please pay more attention in using abbreviations in the entire text. An example of inconsistency: PEL is first introduced in Fig3 and line 231 without explaining! Then in line 318.

> Abbreviations were indeed not accurately introduced and used more carefully in the revised manuscript, in particular Fig.3.

22. Figure 6: would it be possible to add the eddy path trajectory for the indicated eddies in this figure?

> We can provide below an example of an eddy trajectory for one Ierapetra anticyclone (IER1 &2, shown in Fig.10). However this will add extra information to figures already dense. It is important to emphasize that Mediterranean eddies to not significantly drift, the Beta effect being very weak due to small Rossby radius. Algerian anticyclones could be an exception to this statement but not concerned by our study. Eratosthenes and Mersa-Matruh anticyclones are cases of anticyclonic attractors, staying always close to a preferred position (see in particular Barboni et al, 2021, in figures 6a and 10 ). Ierapetra anticyclones also rarely drift significantly far from their formation region, as studied by Ioannou et al (2017).

23. The section 5.5 is suggested to be removed from the ms as i) it is not well discussed ii) out of the focus of the ms.

[Figure]

Ierapetra anticyclone trajectory (IER 1&2)

Section 5.5 was improved to better link with previous discussion. However we would argue it is quite important to keep it. This manuscript focuses indeed on eddy-induced MLD only from a physical point of view, although MLD has a great impact on biological activity. An opening paragraph on the expected biological impact of such eddy modulation then seems very appropriate. Reviewer 1 actually proposed to extent it on Oxygen which is also related to biology.

Besides, it allows to link the findings of this research with previous studies in the Eratosthenes (or Cyprus) anticyclone with more biological focus, in particular Krom et al (1992) and Moutin et Prieur (2012). The later discussed the possibility of winter mixed layer depth not reaching the main nitracline, which is very likely the case in a 'non-connecting' event, while they did not observe it. As our study indeed found this hypothetical event it seems very interesting to keep this paragraph.

**Appendix:**

How does the quality control algorithm work? How does it remove bad quality data? How to you define bad data/spikes in T/S profiles?

Bad quality data (QC flag =1,2 or 5) are removed in the sense they are not kept for interpolation. Profiles filtered from bad data are linearly interpolated on the same vertical vector ( 5m step from 5m to 300m deep, then 10m step from 300m to 2000m deep). After this interpolation, remaining spikes on temperature are removed (replaced by NaN) if there is a jump higher than +6°C or lower than -2°C between 2 points (then dz=5 or 10m). Spikes are corrected only for temperature as this issue almost exclusively concerns noisy XBT.

---

## Referee Report (RR1)

Barboni et al. 2022, Review by A. Capet - 2nd Round.

- **Comment from 1st round** : (Sect 2.1): Mention here already the multi-platform nature of the datasets (Argo, XBT, gliders, etc ..). It would then be relevant to provide the relative abundance of profiles obtained from each platform type. Also mention here explicitly, although briefly, that DT profiles are favored in case of spotted duplicates with NRT datasets.
- **Answer from authors** : *"We were indeed quite brief on the dataset description, as it was intended to be published separately in a data paper, in order to analyze with more details the mean eddy impact on climatology. The following figures should answer this question :"*
  - -> The figures indeed answers the question for me, but it does not answer it for the readers, as no corresponding edit has been made. I believe this can be addressed with one or two additional sentences in (now) Sect. 2.2.

L3 : "heat flux increaseS and restratifieS" or "heat fluxes"

L23 : "**A**egean"

Fig 3 : I think that the black $t^{EA}_{max}$ on the figure's top should be replace with $t^{back}_{max}$.

Fig 8. Consider 'connecting **events'** instead of 'connecting **data'** in the legend.

L 462 / L469 L 297 :  ! homogeneized -> homogenized

---

## Author Response (AR2)

**Barboni et al. 2022, Review by A. Capet**

**Comment from 1st round :** (Sect 2.1): Mention here already the multi-platform nature of the datasets (Argo, XBT, gliders, etc ..). It would then be relevant to provide the relative abundance of profiles obtained from each platform type. Also mention here explicitly, although briefly, that DT profiles are favored in case of spotted duplicates with NRT datasets.

**Answer from authors :** "We were indeed quite brief on the dataset description, as it was intended to be published separately in a data paper, in order to analyze with more details the mean eddy impact on climatology. The following figures should answer this question :"

**Comment from 2nd round :** The figures indeed answers the question for me, but it does not answer it for the readers, as no corresponding edit has been made. I believe this can be addressed with one or two additional sentences in (now) Sect. 2.2.

**2nd answer from author** : 2 sentences were added in Sect.2.2 to describe dataset multi-platform distribution and briefly comment their temporal availability :

'These datasets are multi-platform, gathering in situ vertical measurements from CTDs, XBTs (mostly before 2008), Argo floats (mostly after 2005, with a strong increase after 2012) and gliders (mostly after 2008), enabling an average 10000 profiles available per year from 2011 onwards'

And at the end of the section :

 'Complete database then accounts for 157053 profiles in total, with the following platform distribution : 8596 CTDs, 11375 XBTs, 60019 Argo, 76967 glider profiles and 96 unspecified.'

Also note that platform distribution is slightly different than what was displayed in the 1st review. It is due to the fact that some CTD and XBT profiles were not listed as such, because the type of file digram can be different from 'CT' or 'XB'. At last few profiles remain with an unknown type (for more details, the reader is referred to CMEMS Product User Manual 013-001-B )